# Single-cell analysis of the early *Drosophila* salivary gland reveals that morphogenetic control involves both the induction and exclusion of gene expression programs

Annabel May, Katja Röper 🔴 *

MRC Laboratory of Molecular Biology, Cambridge Biomedical Campus, Cambridge, United Kingdom

* kroeper@mrc-lmb.cam.ac.uk

## Abstract

How tissue shape and therefore function is encoded by the genome remains in many cases unresolved. The tubes of the salivary glands in the *Drosophila* embryo start from simple epithelial placodes, specified through the homeotic factors Scr/Hth/Exd. Previous work indicated that early morphogenetic changes are prepatterned by transcriptional changes, but an exhaustive transcriptional blueprint driving physical changes was lacking. We performed single-cell-RNAseq-analysis of FACS-isolated early placodal cells, making up less than 0.4% of cells within the embryo. Differential expression analysis in comparison to epidermal cells analyzed in parallel generated a repertoire of genes highly upregulated within placodal cells prior to morphogenetic changes. Furthermore, clustering and pseudotime analysis of single-cell-sequencing data identified dynamic expression changes along the morphogenetic timeline. Our dataset provides a comprehensive resource for future studies of a simple but highly conserved morphogenetic process of tube morphogenesis. Unexpectedly, we identified a subset of genes that, although initially expressed in the very early placode, then became selectively excluded from the placode but not the surrounding epidermis, including *hth*, *grainyhead* and *tollo/toll-8*. We show that maintaining *tollo* expression severely compromised the tube morphogenesis. We propose *tollo* is switched off to not interfere with key Tolls/LRRs that are expressed and function in the tube morphogenesis.

## Introduction

During embryonic development complex shapes arise from simple precursor structures. How organ shape and hence function is encoded by the genome remains in many cases an open question. Although we understand often in great detail how gene regulatory networks are specifying overall organ identity, how such patterning is then turned into physical morphogenetic changes that actually shape tissues is much

**Data availability statement:** The datasets and computer code produced in this study are available in the following databases: RNA-Seq data: Gene Expression Omnibus (GSE271294); scRNAseq analysis computer scripts: DOI: 10.5281/zenodo.15011856, and can also be found on GitHub: https://github.com/roeperlab/SalivaryGland_scRNAseq; FACS-data including gating information can be found at: https://doi.org/10.6084/m9.figshare.28625339 Data from single cell analysis can be found in supplemental tables as described in Table 2. Raw data from quantifications of immunofluorescence intensity measurements and apical area quantifications are provided in supplemental data files.

**Funding:** This work was supported by the Medical Research Council, as part of United Kingdom Research and Innovation (also known as UK Research and Innovation; https://www.ukri.org/councils/mrc/) [MRC file reference number MC_UP_1201/11 to KR]. The funders had no role in study design, data collection and analysis, decision to publish, or preparation of the manuscript.

**Competing interests:** The authors have declared that no competing interests exist.

less understood. We use the formation of the tubes of the *Drosophila* embryonic salivary glands as a simple model of tube morphogenesis through budding, a common pathway to form tubular organs [1]. The salivary glands are initially specified at stage 10 of embryogenesis as two flat epithelial placodes of approximately 100 cells each on the ventral side of the embryo. The morphogenesis begins with cells in the dorsal-posterior corner constricting their apices and beginning to internalize. Whilst cells disappear through the invagination point on the surface, a narrow lumen tube forms on the inside [2–5].

Studies over the last 30 years have revealed the transcriptional patterning that leads to the specification of the salivary gland placodes, with the key activator being the homeotic transcription factor Sex combs reduced (Scr; [6,7]). Scr becomes restricted to parasegment 2 in the embryo through the combined action of homeotic transcription factors T-shirt and Abdominal-B repressing *Scr*'s expression posteriorly [8]. Dpp signaling affects the dorsal expansion of the Scr domain [6,8]. Furthermore, classical studies of mutants have revealed many key factors involved in salivary gland morphogenesis [3,9]. The initial primordium, once specified, is quickly subdivided into two groups of cells, secretory cells that will form the main body of the tube, and duct cells, close to the ventral midline (Fig 1A'), that will eventually form a Y-shaped duct connecting both glands to the mouth, once all secretory cells have internalized. These two groups of cells, we know, are established through EGF signaling emanating from the ventral midline and inhibiting Forkhead (Fkh) transcription factor function in the future duct cells (Fig 1A' and 1A''). Fkh in the rest of the placodal cells instructs the morphogenesis and activates a secretory program [10,11]. EGFR mutants do not form a duct and internalize salivary gland tubes with two closed ends [12,13].

In particular Fkh and Huckebein (Hkb) have emerged as transcription factors downstream of Scr that affect key aspects of the morphogenesis of the glands [14,15]. In *fkh* mutants, no invagination or tube forms and Scr-positive cells remain on the surface of the embryo [5,14]. In *hkb* mutants by contrast cells invaginate, though in a central position and the forming glands have highly aberrant shapes [4,15]. Classical studies using embryo sections have suggested that the apical constriction and cell wedging at the forming invagination point is a key part of the internalization of the tube, and that this is preceded by a lengthening of placodal cells and a repositioning of their nuclei towards the basal side [15]. We have previously performed quantitative analyses of the live morphogenetic changes occurring in the early salivary gland placode, and identified two key additive cell behaviors, apical constriction or cell wedging near the forming invagination point as well as directional cell intercalation in cells at a distance to the pit (Fig 1A'). Furthermore, 4D investigation of cell behaviors also indicated further patterned changes with cells tilting and interleaving [5]. The apical constriction, we showed, is driven by a highly dynamic pool of apical-medial actomyosin [16]. We identified that the apical constriction spreads out across the placode in a form of a standing wave, with intercalations feeding more cells towards the pit and cells switching from intercalation to apical constriction once they reach the vicinity of the pit. We could show that these

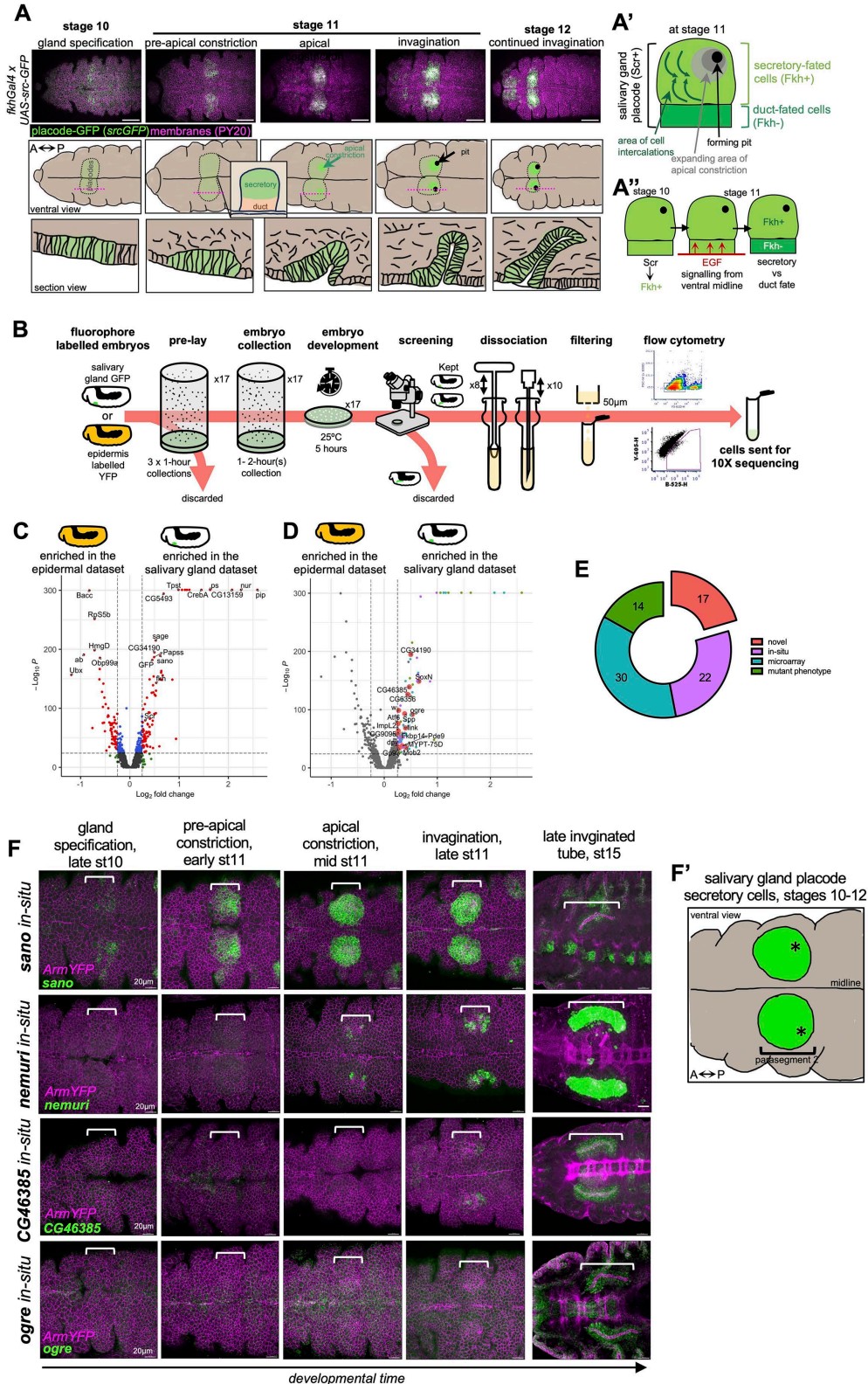

**Fig 1. Generation of a single cell transcriptome dataset of salivary gland placodal and epidermal cells. A)** Immunofluorescence images and matching schematics of the anterior ventral half of *Drosophila* embryos at the indicated stages, highlighting the cells of the salivary gland placode,

labelled with *fkhGal4 x UAS-srcGFP*. All cell outlines are labeled for phosphotyrosine to label adherens junctions (PY20, magenta) and srcGFP is in green, the stage 10 panel is shown with increased exposure, other panels are identical exposure. Scale bar is 100µm. Middle panels are matched schematics, showing the position of the salivary gland placode in pale green, the area of initial apical constriction in bright green and the forming invagination pit in black. Lower panels show schematics of cross sections (positioned indicated by magenta dotted lines in middle panels) of the invaginating tube. **A')** The salivary gland placode becomes split into future secretory and duct cells. Apical constriction near the forming pit and cell intercalations at a distance to the pit drive the cell internalization. **A")** EGF signaling from the ventral midline suppresses Fkh in the future duct cells, whereas in the secretory cells Fkh activates the secretory program and drives the morphogenesis. **B)** Schematic overview of the experimental pipeline for salivary gland placode (*fkhGal4 x UAS-srcGFP*) and epidermal (*armYFP*) cell isolation, FACS and 10X sequencing. **C-E)** Pseudobulk differential expression analysis between salivary gland placodal (*fkhGal4 x UAS-srcGFP*) and epidermal (*armYFP*) cells, confirmatory genes highlighted in (**C**) and upregulated morphogenetic candidates highlighted in (**D**). **E)** Genes specifically upregulated in salivary gland placodal cells were either known to show a phenotype in the glands when mutated (14), had been found to be expressed in the glands by whole-embryo microarray (30), had in situ images on public databases showing gland expression (22) or were completely novel with regards to expression or function in the salivary glands (17), colours are matched between **D** and **E**. **F)** *In situ* by HCR for two novel identified genes upregulated in the salivary gland compared to the epidermis (*CG46385*, *ogre*) as well as for *nemuri* and *sano* that were also highly upregulated in our analysis in **D**. Cell outlines labelled by ArmYFP are in magenta and each respective in situ in green. Shown are representative images of early morphogenetic time points of the salivary gland placode as well as fully invaginated salivary glands. Embryonic stages and morphogenetic descriptors are indicated above the panels. Scale bars are 50µm. Brackets indicate the position of the salivary gland placodes. See also S1 and S2 Figs.

changes are pre-patterned, at least in part, by dynamic transcriptional changes in the expression and protein distribution of the transcription factors Fkh and Hkb [4].

Thus, not only tissue fate but also morphogenetic changes during tube budding are governed by transcriptional changes. Our previous analyses strongly suggest that the cell shape changes and behaviors that drive the tube budding morphogenesis are prepatterned transcriptionally in the tissue. We therefore decided to establish the transcriptional blueprint of the salivary gland placode just prior to and throughout the early stages of its morphogenesis using single cell genomic approaches, a blueprint required to initiate and drive the physical changes. Previous efforts to establish gene expression profiles across salivary gland tube morphogenesis included candidate in situ [17,18] as well as ChIPseq approaches [19,20], whole embryo microarrays [17,18,21] and recently also whole embryo scRNAseq approaches [22–25], though these latter approaches suffered from the fact that salivary gland placodal cells only constitute a very small fraction of all embryonic cells (and thus the datasets), or salivary gland placodal cells were not identified.

Here we generate and utilize a single cell RNA-sequencing dataset of isolated salivary gland placodal cells in comparison to isolated epidermal cells covering the early aspects of salivary gland tube morphogenesis. Pseudo-bulk differential expression analysis identifies a set of novel gland-specific factors, whereas the single cell analysis across pseudotime reveals complex regulatory patterns of expression. To our surprise, not only specific upregulation of factors either across the whole salivary gland primordium or within sections of it occurs, but in addition a number of factors become specifically downregulated and excluded in their expression from the placode, just at the start of morphogenesis. We identify that in certain cases this exclusion of expression is key to wild-type tube morphogenesis. In particular, we show that the ectopic continued expression of one of these factors, the Leucin-Rich Repeat receptor (LRR) Tollo/Toll-8, leads to aberrant morphogenesis, due to Toll-8 interference with endogenous systems, most likely the patterned expression and function of further LRRs, required for correct morphogenesis.

## Results

### Generation of salivary gland and epidermal single-cell RNA sequencing datasets

In order to obtain a single cell RNA-sequencing dataset of salivary gland placodal cells covering the earliest stages from just after specification to early morphogenesis, as well as a matching epidermal cell dataset, we developed a new experimental pipeline: embryos of the genotypes *fkhGal4 x UAS-srcGFP* (for the salivary gland placodal cells; [13]) and *Armadillo/β-Catenin-YFP* (for the epidermal cells) were collected over a 1–2 hours period and aged for 5 hours before being subjected to further visual screening and selection in order to enrich for embryos of the desired stage (Fig 1A and 1B).

The *fkhGal4* driver used is based on a 1kb fragment of the *fkh* enhancer that drives expression very early in the salivary gland primordium, just downstream of specification [26], and in combination with expression of a membrane-targeted form of GFP [13] highlights placodal cells early on (Fig 1A). The Armadillo-YFP fly stock contains a YFP-exon trap insertion into the *arm* locus, thereby labeling the endogenous protein [27]. Embryos were dissociated in a lose fitting Dounce homogenizer and filtered through a 50μm mesh to remove debris before being subjected to flow cytometry to sort GFP- or YFP-positive cells, and these were then subjected to 10X Chromium sequencing (Fig 1B, for details see Materials and Methods). Following quality control steps and setting limits aimed at removal of doublets, dying, and unspecified cells (Materials and Methods; S1 and S2 Figs), a total of 3,452 salivary gland placodal cells and 2,527 epidermal cells were obtained and integrated into a single dataset for downstream analysis.

## Comparison of salivary gland placodal to epidermal gene expression at the onset of salivary gland tubulogenesis

We initially investigated differential expression of genes between the complete salivary gland placodal and epidermal datasets in a pseudo-bulk analysis to, firstly, benchmark and quality control both datasets and, secondly, identify novel upregulated candidates within the salivary gland placodal dataset (Fig 1C–1E; S1 Table). Within the salivary gland dataset, we identified *GFP*, *fkh* and *Scr* as upregulated (*fkh* $Log_2$Fold 0.535539503, p-value 8.50E-144, GFP $Log_2$Fold 0.456845734, p-value 1.02E-182, *Scr* 0.311639192, p-value 2.98E-80), as would be expected from early placodal cells expressing a GFP label. Conversely, the epidermal dataset showed upregulated expression of *abdominal* (*ab*), a gene expressed only posteriorly to location of the salivary gland placode (Fig 1C; Log2Fold −0.9333054, p-value 1.19E-191).

Analysis of the most upregulated genes within the salivary gland placodal in comparison to the epidermal dataset revealed 86 genes (Fig 1D). Of these, only 14 had previously been found to show salivary gland defects when mutants were analyzed [*pipe* [28], *pasilla* [29], *CrebA* [30], *Btk29/Tec29* [31], *PH4alphaSG2* [32], *sage* [33,34], *myospheroid* [35], *fkh* [36], *eyegone* [12,37], *fog* [38,39], *Scr* [40,41], *KDEL-R* [42], *crossveinless-c* [43], *ribbon* [44]], 30 had been described to be expressed within the salivary gland by microarray analysis [*nemuri*, *CG13159*, *Hsc703*, *windbeutel*, *Papss*, *CG14756*, *PH4alphaSG1*, *SsRbeta*, *sallimus*, *TRAM*, *Sec61beta*, *twr*, *piopio*, *Spase12*, *PDI*, *CG7872*, *Surf4*, *Calr*, *CHOp24*, *par-1*, *p24-1*, *Trp1*, *Sec61gamma*, *nuf*, *RpS3A*, *Spase25*, *ERp60*, *Prosap*, *Fas3*, *fili*, [18,19,21]] and 22 could be identified to be expressed in the salivary glands or placode through publicly available in situ hybridization databases [*Tpst*, *CG5493*, *sano*, *CG5885*, *Sec61alpha*, *Tapdelta*, *Gmap*, *l(1)G0320*, *nyo*, *NUCB1*, *Manf*, *bai*, *ergic53*, *CG32276*, *eca*, *GILT1*, *Spase2223*, *Glut4EF*, *CG17271*, *bowl*, *CG9005*, *Tl*; [45–49]]. Seventeen genes were highly upregulated that had not been previously linked to salivary gland morphogenesis or function by any of the above means (*SoxN*, *ogre*, *CG34190*, *CG46385*, *CG6356*, *Mob2*, *link*, *Spp*, *MYPT-75D*, *Pde9*, *Fkbp14*, *Gp93*, *w*, *ImpL2*, *CG9095*, *Atf6*, *dpy*). To further validate these genes identified as salivary gland expressed, we performed in situ hybridization using hybridization chain reaction (HCR) for mRNAs of several genes that were either novel or where no spatio-temporal expression data existed, including *nemuri* that only began to be expressed in the salivary gland placode during onset of apical constriction, and *sano*, that began expression very early on in a region prefiguring the position of the forming invagination pit (Fig 1F).

Thus, this differential expression analysis confirmed that our cell isolation and sequencing method was able to generate high-quality datasets for further in depth analyses, and also revealed that we could identify novel expression of genes across a spread of early stages of salivary gland specification and morphogenesis.

## Generation of a salivary gland cell atlas by single-cell RNAseq

We now focused on the salivary gland lineage in the dataset and used uniform manifold projection to identify clusters of cells with related expression profiles across this dataset. At a resolution of 0.17 the data split into 8 clusters (Fig 2A; S2 Table). Analysis of top expressed genes predicted that these represented epidermal cells not yet specified to become salivary gland, salivary gland cells, anterior midgut cells, CNS cells, Enhancer of split [E(spl)]-enriched cells, amnioserosa cells, muscle cells, and hemocytes. The presence of cells other than salivary gland placode cells in this dataset was most

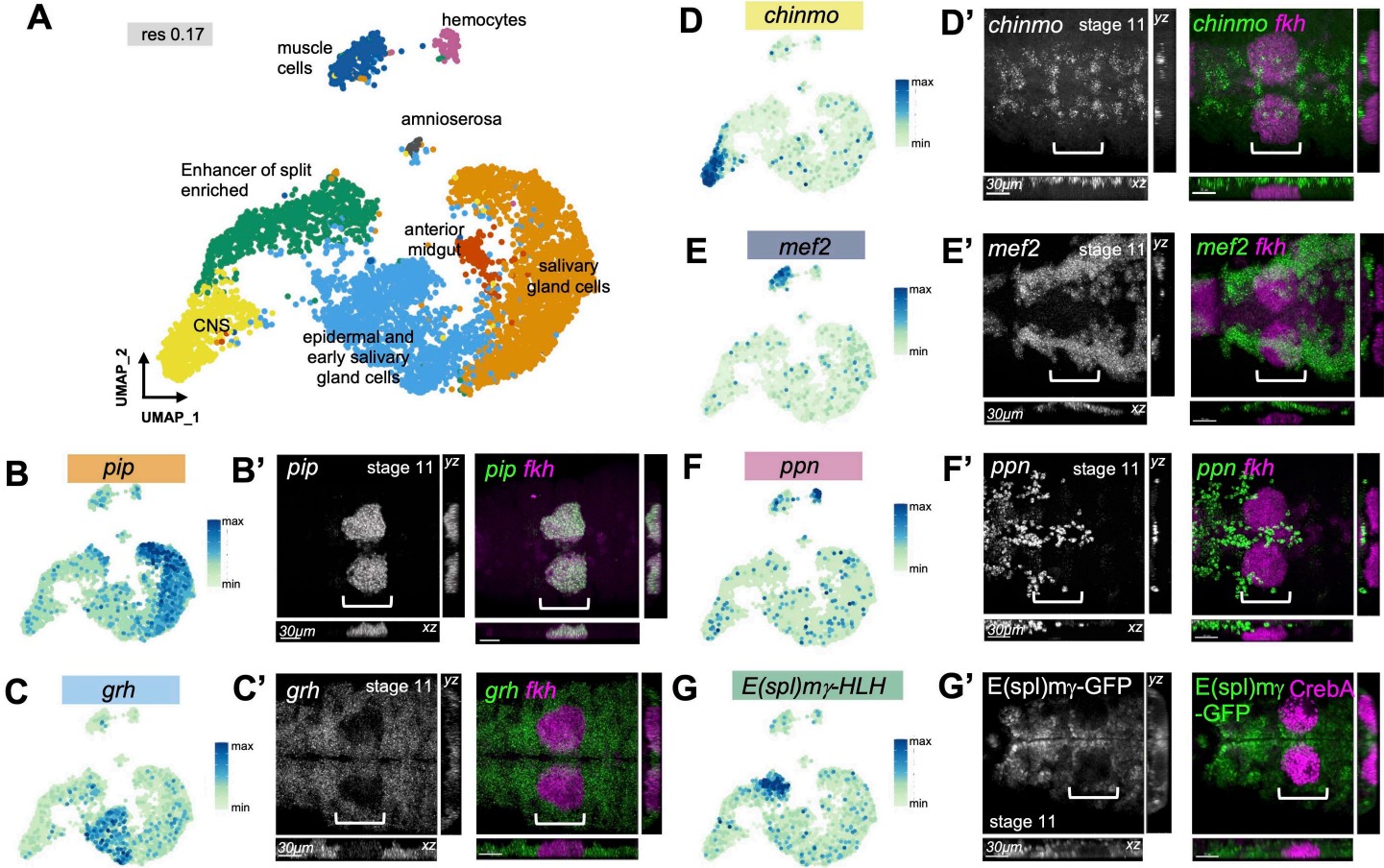

**Fig 2. Single cell atlas of gene expression across early salivary gland morphogenesis. A)** Combined UMAP plot of all cells isolated in the *fkhGal4 x UAS-srcGFP* and *ArmYFP* samples, clustered at resolution 0.17. Salivary-gland-specific and nonspecific clusters are identified. **B-G')** Expression of upregulated marker genes for each cluster plotted onto the UMAP **(B-G)**, with in situ hybridization images by HCR for each gene shown in **B'-F'**, and an endogenous GFP-protein trap showing cells expressing E(spl)mγ-GFP (**G'**). Marker genes are shown as individual channels and in green in merged channels. Images are of mid stage 11 embryos at a point when apical constriction has commenced. Images show faces on views with matching xz and yz cross-sectional views, to distinguish labeling at different depths. White brackets indicate the position of the salivary gland placode, identified by either *fkh* mRNA expression (**B'-F'**) or CrebA staining (**G'**). Scale bars are 30μm. See also S3 Fig.

likely due to the low-level expression of the *fkhGal4* line also outside the salivary gland placode (see Figs 1A and S1A), and also possible contamination due to mechanical dissociation applied to isolate cells and gating regime during FACS, as we aimed to isolate as many of the low number of salivary gland cells in the embryo as possible. We aimed to confirm the identity of clusters and the expression or absence as well as localization of expression of top markers for each cluster identified (S2 Table) in the salivary gland placode and surrounding epidermis at stages when the morphogenesis had clearly commenced, i.e., 'apical constriction' at mid stage 11, and performed HCR *in situs* for these.

  *pip* (*pipe*), a sulfotransferase of the Golgi is key to the later secretion function of the salivary glands and a known marker of these as analyzed previously [19], and its transcript colocalized already early on with *fkh* mRNA, confirming this cluster as 'salivary gland' (Fig 2B and 2B'). The top marker gene for the epidermal/ early salivary gland cell cluster, *grh* (*grainyhead*), encoding a pioneer transcription factor [50–52] was expressed throughout the epidermis when analyzed at stage 11 when apical constriction had commenced, not overlapping with *fkh* mRNA (Fig 2C and 2C'). Further analysis of this cluster through increasing the clustering resolution confirmed the presence of secretory and ductal salivary gland as

well as a range of epidermal markers expressed in cells of this cluster (S3 Fig). *chinmo*, encoding a transcription factor with a key timing role in the nervous system [53,54], localized to groups of neuronal precursors, and although some of the expression looked to overlap *fkh* mRNA, 3D analysis revealed that *chinmo* expressing cells were in fact localized further interior than the salivary gland cells (Fig 2D and 2D'). *mef2*, encoding a muscle transcription factor [55], was expressed in muscle precursors at stage 11, localizing further interior than *fkh* expressing placodal cells (Fig 2E and 2E'). *ppn* (*papilin*) encodes a component of the extracellular matrix (ECM) [56] that is, as is most embryonic ECM, expressed by hemocytes during their embryonic migration, and its expression therefore did not colocalize with *fkh* mRNA (Fig 2F and 2F'). *E(spl)* gene expression dominated one cluster [57,58], and analysis of a GFP-trap in E(spl)mγ at stage 11 revealed protein expression across the epidermis but excluded from the salivary gland placode marked by CrebA immunostaining at this stage (Fig 2G and 2G').

Thus, the single cell RNA-sequencing analysis of combined ArmYFP-labelled and enriched placodal cells (using *UAS-srcGFP fkhGal4*) was able to generate a cell atlas of the salivary gland placode as well as its precursor epidermis and nearby tissues at early stages of embryogenesis that provides a rich resource of expression data for these stages.

**Salivary gland specific clusters reveal temporally controlled expression of many factors potentially affecting morphogenesis**

Increasing the resolution of clustering and homing in on salivary gland-related cells revealed a split into 4 clusters (Figs 3A and S4A–S4F). Analysis of the highest expressed genes for each of these clusters allowed us to order them in a predicted temporal progression representing aspects of salivary gland morphogenesis: 'early salivary gland', 'specified secretory salivary gland cells', 'specified duct salivary gland cells', 'post specification/late salivary gland cells' (Fig 3A and 3B; S3 Table). As discussed above, early specified salivary gland placodal cells are committed to either secretory or duct cell lineage by EGFR sigaling from the ventral midline (Fig 1A' and 1A''; [12]). To understand better what set each cluster apart from the others, we looked at differential gene expression for each of these clusters (Fig 3B and 3C). Each cluster displayed a selective upregulation of genes, some of which had previously been linked to salivary gland morphogenesis or function, and others not implicated or known to be expressed in the placode or glands. We therefore performed HCR in situ hybridization to confirm the temporal changes in expression of marker genes for each cluster along the salivary gland morphogenesis trajectory predicted from the increased clustering, especially focusing on early stages of the process, split into 'early placode', 'pre-apical constriction', 'apical constriction' and 'continued invagination' (Fig 3D). As previously described, *hth*, encoding a transcription factor working in conjunction with Scr and Exd in salivary gland placode specification [6], was expressed very early in the primordium and then appeared to be downregulated and actively excluded from the placodal cells (Figs 3D and S4B). Uncharacterized gene *CG45,263*, a top marker gene within the specified duct cell cluster (Fig 3B and 3C) was expressed in a spatial pattern that initiated close to the ventral midline with expansion into the duct cells of the salivary gland primordium during pre-apical constriction stages. Similar to the expression timing previously reported for components related to EGFR signaling within the duct cells of the salivary gland primordium [26], expression of *CG45263* within the duct portion of the salivary glands continued even post-invagination (Figs 3D and S4C). As previously described, mRNA for Gmap, a Golgi-microtubule associated protein, was expressed within the salivary gland placode beginning at early stage 11 [59]. The in situ hybridization covering early placodal development confirmed this observation and furthermore revealed that the expression originated at late stage 10 in the cells first to invaginate, but that the onset of *Gmap* expression extending to all secretory cells was later than the onset of *fkh* expression (Fig 3D; S4D Fig versus S4A Fig). *calreticulin* (*calr*), the top marker gene of the 'post specification' cell cluster has not previously been implicated in a specific salivary gland development function. It showed a later onset of expression in the salivary gland placode than both *fkh* and *Gmap*, displaying a diffuse expression beginning during pre-apical constriction in all secretory cells with expression increasing as invagination continued (Figs 3D and S4E).

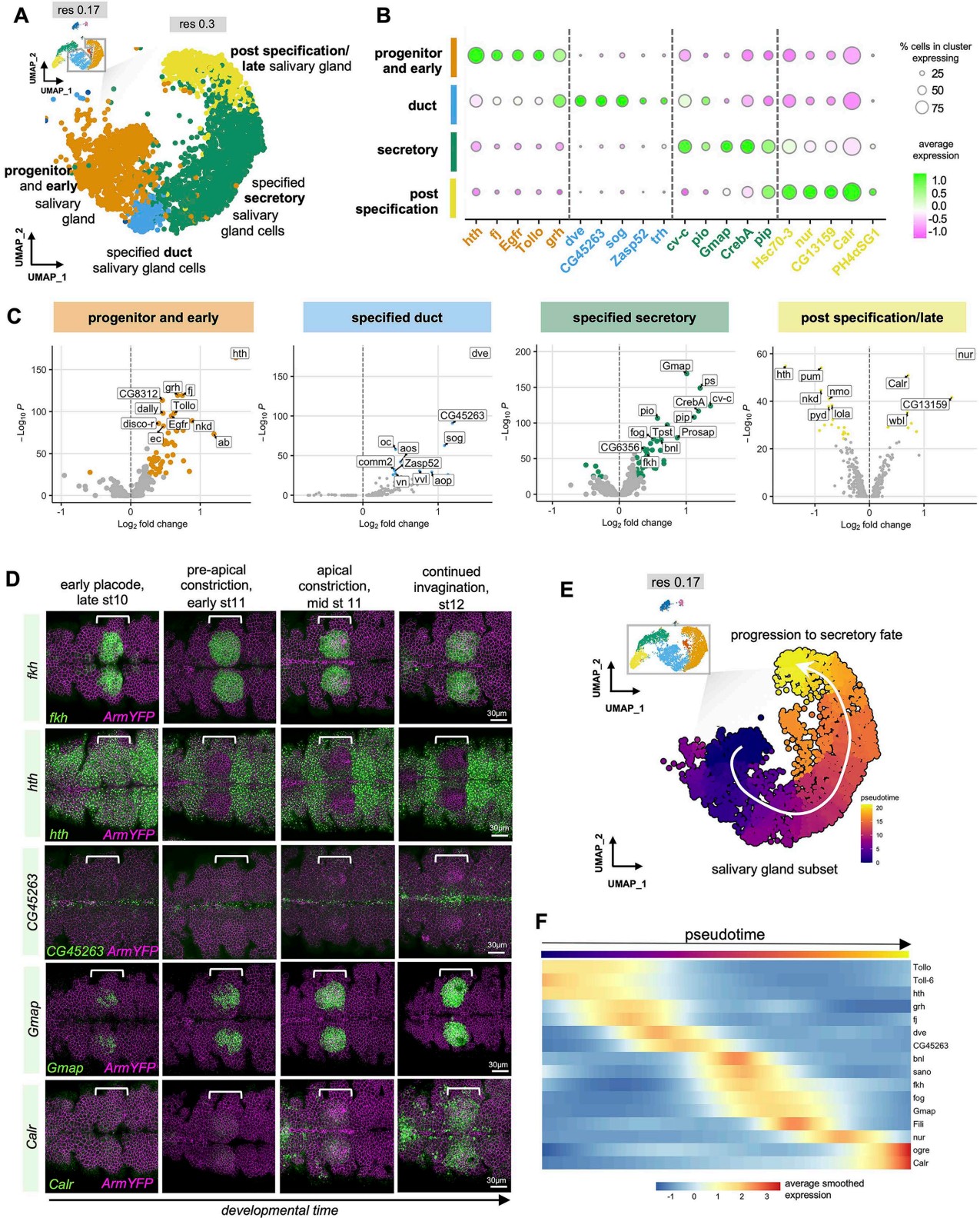

**Fig 3. A proposed single cell timeline of mRNA expression changes during salivary gland morphogenesis. A)** Higher resolution UMAP of the salivary gland lineage subsection derived from the combined data sets (resolution 0.3), with clusters indicating a likely temporal progression along

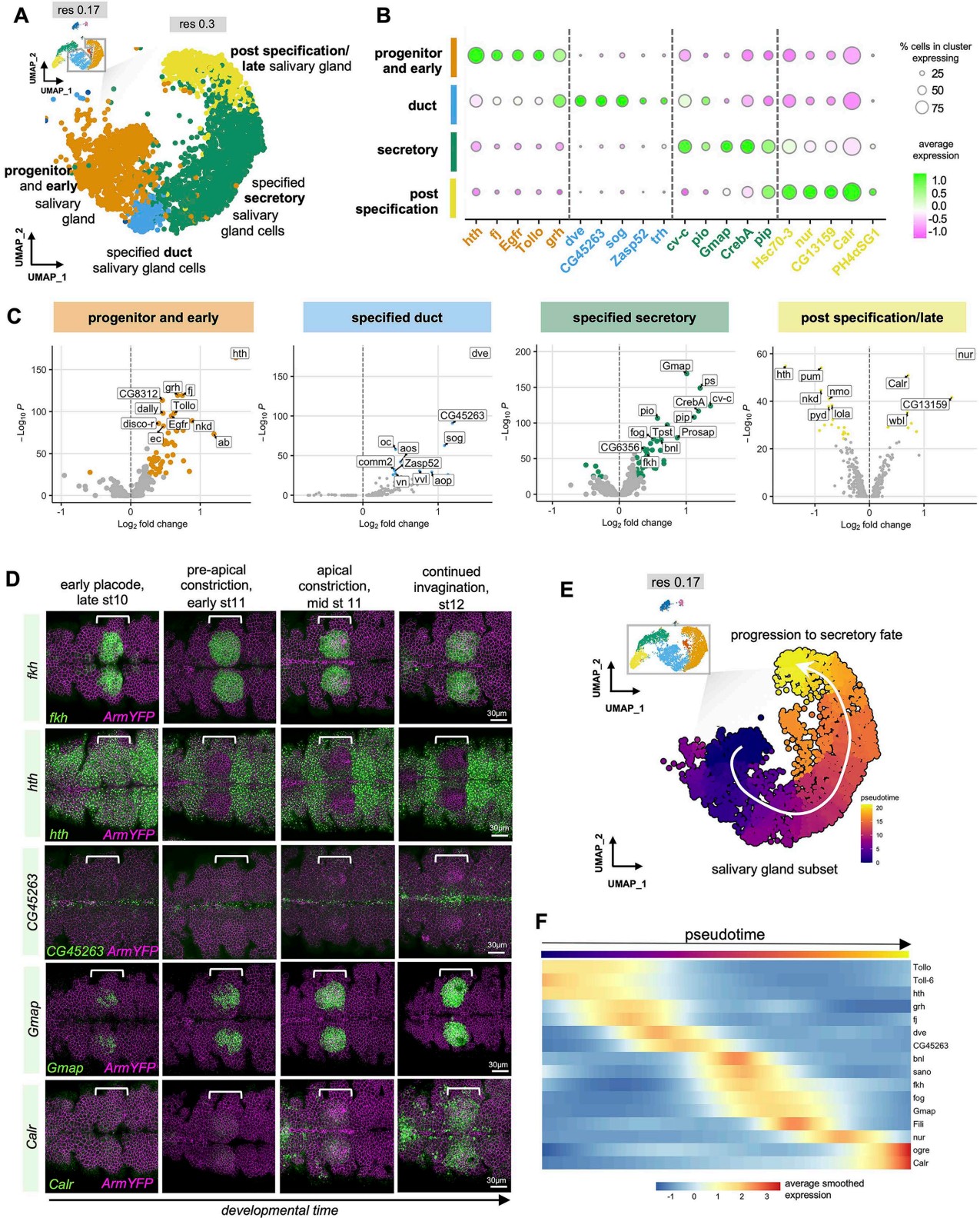

salivary gland morphogenesis labeled. **B)** Expression analysis of the top five marker genes of each cluster identified in **A**: progenitor and early salivary gland, specified duct cells, specified secretory cells and post specification/late salivary gland cells. Color indicates level of expression and the size of the circle indicates the percent of cells in each cluster expressing the gene, vertical dotted lines denote cluster boundaries. **C)** Volcano plots displaying upregulated marker genes in each of the clusters identified in **A**, each cluster is compared to the remaining cells in the dataset. **D)** In situ hybridization by HCR of one top marker gene per cluster identified in comparison to *fkh* expression: *hth* for the 'progenitor and early gland' cluster, *CG45,263* for the 'specified duct cells' cluster, *Gmap* for the 'specified secretory cells' cluster and *Calr* for the 'post specification/late' cluster. White brackets indicate the position of the salivary gland placodes, scale bars are 30µm, in situ for marker genes is in green, ArmYFP to label cell outlines is in magenta. **E** Pseudotime analysis based on the proposed salivary gland portion of the lower resolution UMAP in Fig 3A. The pseudotime agrees with a lineage progression to secretory fate as one possible trajectory. **F)** Based on the pseudotime trajectory a subset of genes was plotted along pseudotime to reveal differential temporal expression along the early morphogenetic timeline (see See also S4 Fig).

Thus, our cluster analysis using increased resolution strongly suggested a temporally controlled pattern of gene expression along the morphogenetic trajectory, in line with previous studies. We now employed pseudotime analysis on the lower resolution cluster to analyze whether this approach would confirm our above analysis. Without specifying origin or endpoint clusters, unsupervised pseudotime analysis using Monocle3 identified a lineage originating from cells previously clustered in the 'early salivary gland' cluster moving towards the 'post specification' cell cluster (Fig 3E) or towards the specified duct cell cluster (S4G Fig). Focusing on these lineages, plotting the assigned pseudotime values of each cell on the previously generated UMAP indicated that cells analyzed in this study are clustered along temporal axes (Figs 3E and S4G). This further validated the cluster assignment of cells in the salivary gland placode from early specification through to post-specification (Fig 3A). Furthermore, the continuous change over time as indicated by the pseudotime analysis matched closely the biological reality of salivary gland tubulogenesis as a continuous process of invagination as opposed to discrete stages. Following the pseudotime trajectory also allowed us to identify and plot the differential expression of 207 genes, some of which are highlighted in Fig 3F (S4 Table).

In summary, the expression analysis of placodal cells at the single cell level revealed an intriguing dynamicity of gene expression activation and cessation across the short time period of salivary gland morphogenesis, suggesting a tight temporal expression control of morphogenetic effectors.

## A cluster of genes with specific exclusion of expression in the salivary gland placode

In addition to the temporally controlled onset of expression of many factors within the salivary gland primordium, we also identified two groups of genes that showed a striking cessation and exclusion of expression during the stages spanning the tube morphogenesis. All of these genes, though, were strongly expressed in the salivary gland primordium early on during or just after specification (Figs 4 and S5).

The first group of genes comprised factors related to gland specification (*hth*) or, as previously described in other tissues, related to epithelial features (*grainy head* [*grh*], *four-jointed* [*fj*], *tollo*). Within the cell cluster defined above in the UMAP plot as containing cells of the early salivary gland (Figs 3A and S3 and S5 and S5 Table), these four were strongly expressed at the point of specification, but switched to being downregulated or excluded by the stage that morphogenesis commenced with apical constriction of cells to form the invagination pit (Fig 4A). In situ hybridzation for *hth*, *grh*, *fj* and *toll-8/tollo* revealed a mutually exclusive pattern of expression when compared to in situ labeling for *fkh* at the stage of apical constriction (Fig 4B). In all cases the exclusion of expression was confined to the future secretory but not the future duct cells in the primordium (compared to *fkh* expression analyzed in parallel).

The second group of genes all belonged to the Enhancer of Split [E(spl)] cluster, a group of transcriptional repressors involved in restricting neurogenic potential downstream of Notch in many tissues [57,58]. *E(spl)m5-HLH*, *E(spl)m4-BFM*, *E(spl)mα-BFM*, *E(spl)m3-HLH*, *E(spl)mγ-HLH*, *E(spl)mδ-HLH* and *BobA* were all identified in this cluster (Figs 4C and S5). Also for this cluster, in situ hybridization or use of GFP-reporter lines revealed and initial expression at stage 10 during specification in the position of the forming salivary gland placode that then quickly resolved into a mutually exclusive pattern of expression when compared to in situ labeling for *fkh* or staining for CrebA, with the downregulation and exclusion

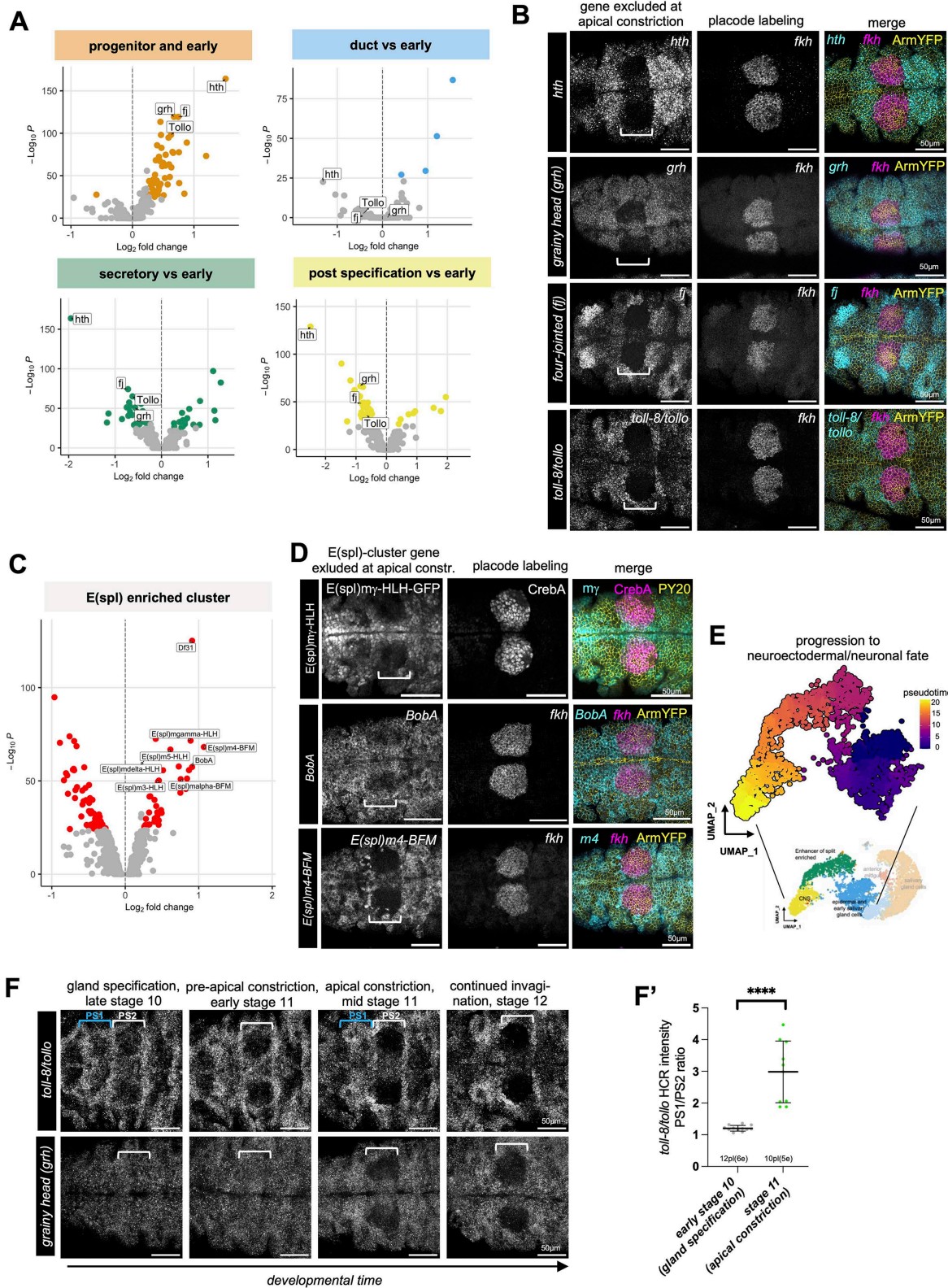

**Fig 4. Placode-specific downregulation and exclusion of expression of candidates. A)** Volcano plots showing candidates upregulated early during gland specification that becomes specifically downregulated in their expression once morphogenesis commences, the 'progenitor and early' plot is

repeated from Fig 3C, and the other three plots compare the indicated datasets to this 'progenitor and early' cluster. Cut-off values are indicated by a change in color. **B)** In situ hybridization by HCR of salivary gland placodes for 4 downregulated genes at apical constriction stage (mid stage 11, *hth*, *grh*, *fj* or *tollo* probes in cyan), in comparison to *fkh* in situ to mark the position of the secretory cells of the placode (magenta) and apical cell outlines showing the apical constriction (ArmYFP in yellow). **C)** Volcano plot showing gene expression enriched in the E(spl)-cluster compared to the rest of the dataset. E(spl) genes are also upregulated early during gland specification but then specifically downregulated and excluded during morphogenesis. Cut-off values are indicated by a change in color. **D)** Endogenously-tagged protein or in situ hybridization by HCR of salivary gland placodes for 3 E(Spl) cluster genes (mid stage 11, E(spl)mγ-HLH-GFP, *BobA* probe and *E(spl)m4-BFM* probe in cyan), in comparison to placode secretory cell labeling via anti CrebA antibody or *fkh* in situ (magenta) and apical cell outlines showing apical constriction (ArmYFP in yellow). **E)** Pseudotime analysis based on the proposed mainly nonsalivary gland portion of the lower resolution UMAP in Fig 3A. The pseudotime outlines a lineage progression from epidermal to neuroecto-dermal and neuronal as one possible trajectory. **F** Timeline of *toll-8/tollo* (top panels) and *grh* (lower panels) downregulation and exclusion of expression in the salivary gland placode compared to the surrounding epidermis. Scale bars are 50 μm, white brackets indicate the position of the placodes in parasegment 2, blue brackets indicate the position of parasegment 1. **F'** Quantification of *toll-8/tollo* downregulation in the secretory cells of the salivary gland placode comparing the gland specification stage (late stage 10; 12 placodes in 6 embryos were analyzed) and the apical constriction stage (mid stage 11; 10 placodes in 5 embryos were analyzed), calculated as the ratio between the intensity per area for an area in parasegment 1, where intensity does not change across this period, to the secretory cell area of the placode. Shown are mean +/- SD, difference was determined as significant by unpaired *t* test as p < 0.0001 (****). See S1 and S2 Data file. See also S5 Fig.

restricted to all or part of the future secretory cells (Figs 4D, S5, S5B and S5C). The pseudotime analysis suggests a potential trajectory of early epidermal cells via the E(spl) cluster towards a neuroectodermal and then neural fate (Fig 4E).

For *toll-8/tollo* and *grh*, two genes encoding factors previously implicated in either epithelial morphogenesis (*tollo*; [60–62]) or control of the epithelial phenotype and characteristics (*grh*; [50–52]), we analyzed the spatio-temporal evolution of transcription for both over the time period of placode specification and early morphogenesis (Fig 4F and 4F'; see also Fig 6 for *toll-8/tollo*). At the gland specification stage, both *toll-8/tollo* and *grh* were still expressed in parasegment 2 where the salivary gland placode will form, but both were clearly excluded from the secretory part of the placode once apical constriction commenced. The same dynamics of downregulation and exclusion of expression were observed for *E(spl)m4-BFM* (S5C Fig).

Thus in addition to the specific upregulation of factors within the salivary gland placode across specification and tube morphogenesis, it appears that there is a concomitant specific downregulation and exclusion of expression of another set of factors, and this exclusion could represent another key part of the transcriptional program driving tubulogenesis.

### Continued placodal expression of Toll-8/Tollo reveals the requirement for its exclusion for correct salivary gland morphogenesis

The specific loss of *toll-8/tollo* expression from the salivary gland placode coinciding with the onset of morphogenetic changes suggested that continued expression of *toll-8/tollo* might interfere with these changes. We therefore decided to re-express Toll-8/Tollo, a transmembrane protein receptor of the leucine-rich repeat (LRR)-family [63,64], specifically in the salivary gland placode under *fkhGal4* control. We initially used expression of a full-length tagged version of Toll-8/Tollo (*UAS-TolloFL-GFP*). In control placodes (*fkhGal4*; Fig 5B, bottom panels) with the start of apical constriction a narrow lumen tube started to invaginate and extend over time whilst cells internalized from the surface. By comparison when Tollo remained present across the placode (in *fkhGal4 x UAS-TolloFL-GFP* embryos), apical constriction appeared disorganized and spread to more cells. Already at early stages *fkhGal4 x UAS-TolloFL-GFP* embryos often showed the invagination of cells at multiple sites across the placode (Fig 5B, arrows in cross sections), rather than the single wild-type invagination point. The invaginations then progressed to a too-wide and misshapen tube, and fully invaginated glands at stage 15 showed highly misshapen lumens and overall shape (Fig 5B). Interestingly, very similar phenotypes were observed when a tagged version of Toll-8/Tollo lacking the intracellular cytoplasmic domain was expressed (*fkhGal4 x UAS-TolloΔcyto-GFP; S4A and S4A'* Fig). In fact, compared to control placodes, *fkhGal4 x UAS-TolloFL-GFP* and *fkhGal4 x UAS-TolloΔcyto-GFP* placodes and invaginated tubes at stage 11 and 12 consistently showed multiple invagination

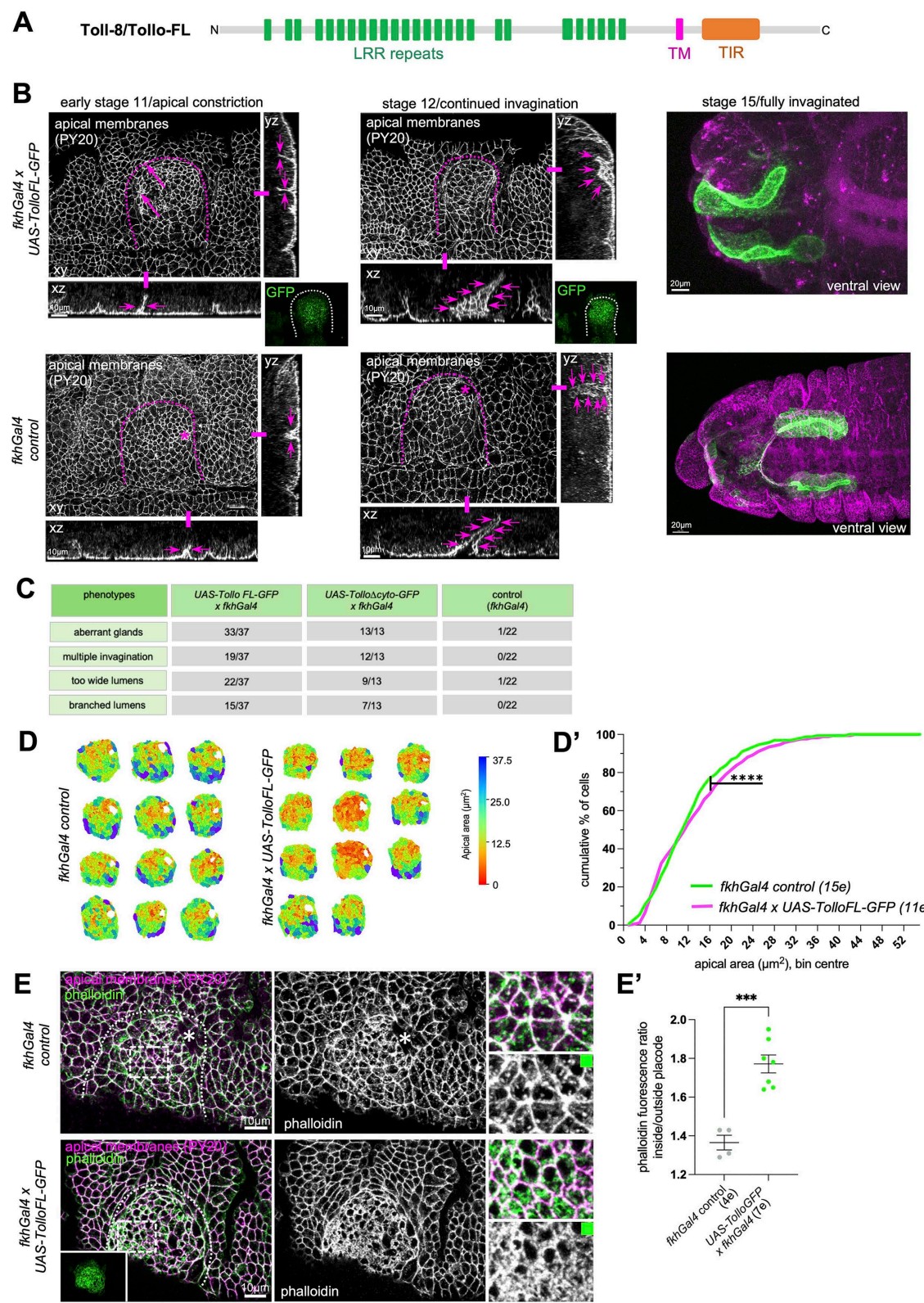

Fig 5. Continued expression of Toll-8/Tollo disrupts salivary gland tubulogenesis. A) Schematic of Toll-8/Tollo full-length (FL) used for re-expression of Toll-8/Tollo in the salivary gland placode using the UAS/Gal4 system, with extracellular leucine-rich repeat (LRR) domains, a

transmembrane domain and an intracellular Toll-interleukin 1 receptor (TIR) domain. **B)** In contrast to control (*fkhGal4 control*, bottom panels) placodes where apical constriction begins in the dorsal posterior corner and a narrow lumen single tube invaginates from stage 11 onwards, in embryos continuously expressing *UAS-TolloFL-GFP* under *fkhGal4* control (top panels) multiple initial invagination sites and lumens form and early invaginated tubes show too wide and aberrant lumens (magenta arrows in cross-section views). Fully invaginated glands at stage 15 show highly aberrant lumens. Apical membranes are labeled with an antibody against phosphotyrosine (PY20) labeling apical junctions. Dotted lines mark the boundary of the placode, asterisks the wild-type invagination point. Green panels show the expression domain of TolloFL-GFP. Scale bars are 10 μm or 20 μm as indicated. **C)** Quantification of occurrence of aberrant glands, multiple invaginations, too wide lumens or branched lumens in either *fkhGal4 x UAS-TolloFL-GFP* or *fkhGal4 x UAS-TolloΔcyto-GFP* compared to *fkhGal4 control*. **D, D')** Quantification of apical area distribution of placodal cells in control (*fkhGal4 control*) and *fkhGal4 x UAS-TolloFL-GFP* placodes when invagination has commenced at stage 11. **D)** Placode examples showing apical area. **D')** Quantification of the cumulative percentage of cells in different size-bins [Wilcoxon matched-pairs signed rank test, p < 0.0001 (****)]. Twelve placodes were segmented and analyzed for control and 11 for *UAS-TolloFL-GFP* overexpression, the total number of cells traced was 1,686 for control embryos and 1,599 for *UAS-TolloFL-GFP* overexpression. See S3 and S4 Data files. **E, E')** Analysis of apical F-actin in placodes of *fkhGal4 control* in comparison to *fkhGal4 x UAS-TolloFL-GFP* labeled with phalloidin. Apical membranes are labelled for phosphotyrosine (PY20). Dotted lines mark the boundary of the placode, asterisks the wild-type invagination point. Green panel shows the expression domain of TolloFL-GFP. Scale bars are 10 μm. **E'** Quantification of apical phalloidin as a ratio of inside to outside placode in *fkhGal4 control* (4 embryos) and *fkhGal4 x UAS-TolloFL-GFP* embryos (7 embryos). Shown are mean +/- SD, difference was determined as significant by unpaired t *test* as p < 0.0001 (****). See S5 and S6 Data files. See also S6 Fig.

points as well as too wide and branched lumens (Fig 5C), suggesting that the intracellular domain of Toll-8/Tollo was not required for its dominant-negative effect when re-expressed in the salivary gland placode.

To address the aberrant apical constriction in a quantitative manner, we segmented the apical area of placodes in *fkhGal4* control and *fkhGal4 x UAS-TolloFL-GFP* embryos after apical constriction had commenced (Fig 5D and 5D'). This revealed an increase in apically constricted cells at stage 11 in embryos where Toll-8/Tollo continued to be expressed in the placode. We reasoned that, firstly, as we had previously shown that apical constriction depends on apical actomyosin in placodal cells [16], and, secondly, Toll/LRR proteins have been implicated in numerous contexts in the regulation of actomyosin accumulation at junctions [60–62,65,66], the aberrant apical constriction could be due to changes in apical actomyosin. In *fkhGal4 x UAS-TolloFL-GFP* embryos, in comparison to *fkhGal4* control, apical F-actin was significantly enriched, in particular at junctions but also extending across large parts of the apical surface (Fig 5E and 5E').

Thus, a continued presence of Toll-8/Tollo lead to significantly disrupted tubulogenesis of the salivary glands, strongly suggesting that the observed expression-exclusion of *tollo* is a key aspect of wild-type morphogenesis.

## Continued placodal presence of Toll-8/Tollo interferes with endogenous LRR function in the placode

Why is the exclusion of *toll-8/tollo* expression important for wild-type tube morphogenesis of the salivary glands? At earlier stages of morphogenesis during gastrulation three Tolls, Toll-2/18-wheeler, Toll-6 and Toll-8/Tollo, show an intricate striped expression pattern repeated every parasegments across the epidermis of the embryo that is key to germband extension movements [61]. Furthermore, mutants in *toll-2/18-wheeler* (*18w*) have previously been reported to show defects in late salivary glands, a phenotype enhanced by further changes in components affecting the Rho-Rok-myosin activation pathway [43]. We therefore analyzed the expression of *toll-2/18w* and *toll-6* in comparison to *toll-8/tollo* across early stages of salivary gland morphogenesis in the embryo (Fig 6A and 6B). At early stage 10, just at the onset of salivary gland placode specification and when *fkh* expression just starts to spread across the placode, all three genes were still expressed in a stripe pattern reminiscent of the earlier striped expression during gastrulation (Fig 6A). At late stage 10/early stage 11, when the salivary gland placode was specified and morphogenesis was about to commence, *toll-2/18w* was expressed at the future invagination point in the placode, with expression radiating out form here, similar to what had been described previously [43]. By contrast, both *toll-6* and, as shown above, *toll-8/tollo* were now, only about 30 min onwards in development, excluded in their expression from the secretory cells of the salivary gland placode (Figs 6B and S7A). All three genes also showed more complex expression patterns in the epidermis surrounding the salivary gland placode at this stage. This change in pattern of placodal expression for these *toll* genes was in agreement with a specific role for Toll-2/18w in placode morphogenesis and a specific downregulation or exclusion of expression of the other previously expressed Tolls, Toll-6 and Toll-8/Tollo. Further supporting the role of

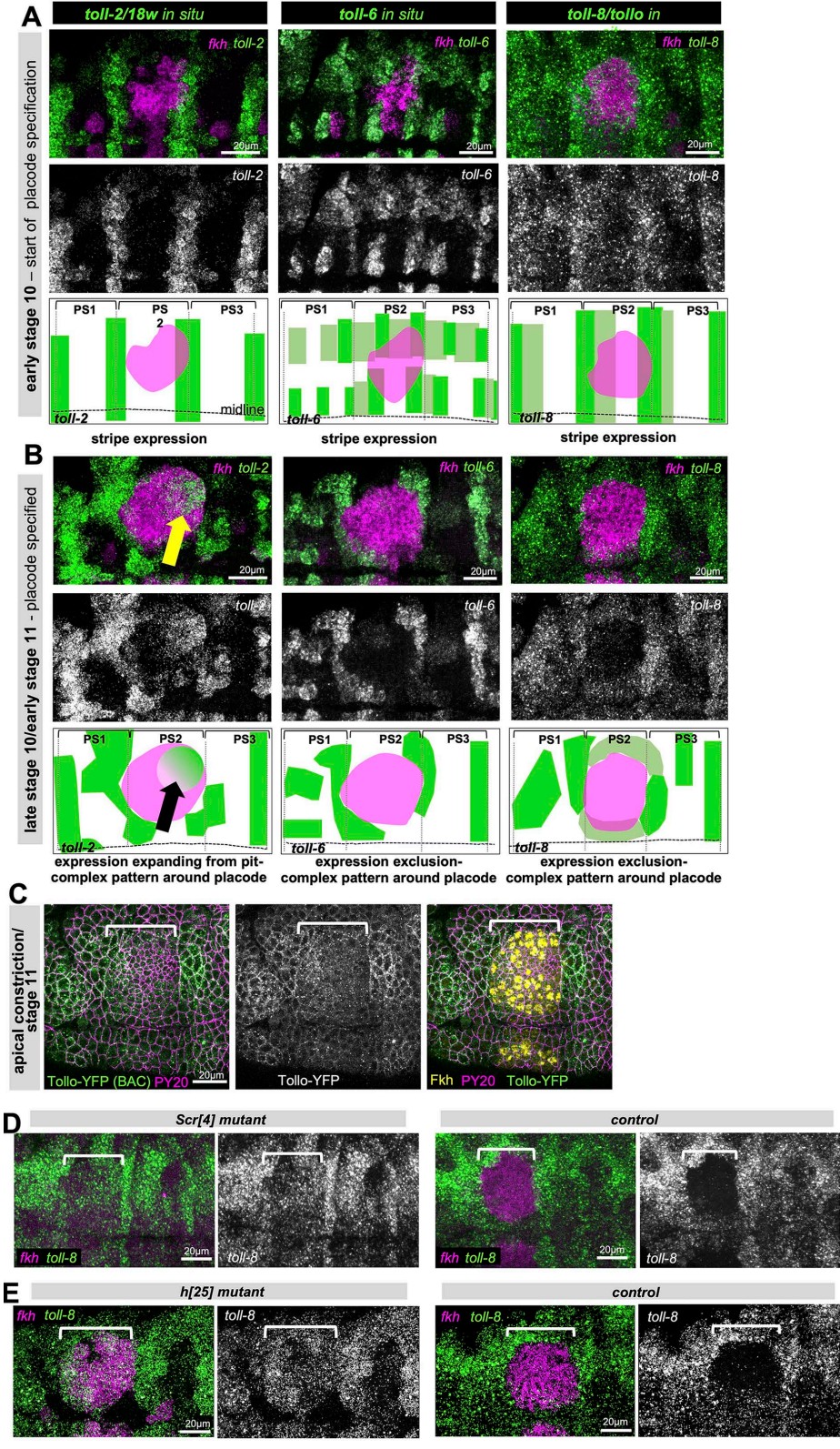

**Fig 6. Continued expression of Toll-8/Tollo disrupts an endogenous LRR code required for proper morphogenesis. A)** At early stage 10, at the very onset of specification of the salivary gland primordium, *toll-2/18w*, *toll-6* and *toll-8/tollo* are still expressed in complementary stripe patterns across

the epidermis. Top row shows in situ hybridizations by HCR for each Toll in comparison to *fkh* in situ, lower panels show *toll-2*, *toll-6* and *toll-8* alone. The schematics, drawn from the immunofluorescence images above, show the stripe expression for each gene and for *fkh* in comparison to the position of the parasegmental boundaries. **B)** At late stage 10/early stage 11 when apical constriction is about to commence, *toll-2*, *toll-6* and *toll-8* show complex expression patterns with *toll-2* specifically expressed around the forming invagination point (yellow arrow) and *toll-6* and *toll-8* specifically downregulated and excluded from the secretory cells of the alivary gland placode. The schematics show the stripe altered expression pattern, drawn from the immunofluorescence images above, for each gene and for *fkh* in comparison to the position of the parasegmental boundaries. **C)** Tollo-YFP (*P{Tollo. SYFP2}*) protein expression (in green) at the apical constriction stage, apical cell outlines are marked using PY20 (anti-phospho-tyrosine) in magenta and placodal cells are marked by Fkh protein in yellow. **D)** Comparison of *toll-8*/*tollo* expression analyzed by in situ (HCR) in *Scr[4]* mutant embryos and control embryos at mid stage 11. *toll-8*/*tollo* is in green and *fkh* in magenta (note the expected absence of *fkh* in the *Scr[4]* mutant). **E)** Comparison of *toll-8*/*tollo* expression analyzed by in situ (HCR) in *hairy* (*h[25]*) mutant embryos and control embryos at mid stage 11. *toll-8*/*tollo* is in green and *fkh* in magenta. Scale bars are 20μm, white brackets in **C-E** show the position of the salivary gland placode. See also S7 Fig.

a dynamic spatio-temporal patterning of expression of Toll-2/18w, that is involved in driving the correct cell behaviors during cell internalization [5,43] and that requires absence of other Tolls that could interact and interfere with Toll2/18w, overexpression of Toll-2/18w itself in the placode (as reported previously; [43]) and overexpression of Toll-6 also led to aberrant patterns of invagination and glands with aberrant shapes and lumens at later stages (S6 Fig). The overexpression effect of Toll-6 and Toll-8/Tollo we suggest are mediated through interaction of the overexpressed proteins with endogenous interactors, most likely Toll-2, as the intracellular domain of Toll-8/Tollo is not required for the overexpression phenotype (S6 Fig). Furthermore, the overexpression effect of Toll-8/Tollo is not mediated by changes in expression of either *toll-2/18w* or *toll-6*, as both show wild-type expression levels and pattern of mRNA under Toll-8/Tollo overexpression (S7B Fig).

We used BAC-mediated transgene expression of Toll-8/Tollo (Tollo-YFP) to analyze Toll-8/Tollo protein expression at early stages of salivary gland morphogenesis, with the caveat that this transgene in a wild-type background, though under control of endogenous elements, nonetheless leads to large fluorescent protein aggregates within the cells, likely due to the fact that cells now contain three copies of the *tollo* gene. We therefore focused just on the apical surface of the placodal and epidermal cells at stage 11 when apical constriction had just begun. Toll-8/Tollo is usually localized to the apical junctional area [62]. At early stage 11, Tollo-YFP appeared to be reduced in its levels within the salivary gland placode compared to the surrounding epidermis, matching its exclusion at the mRNA level (Fig 6C).

Lastly, we wanted to test whether the specific downregulation and exclusion of *toll-8/tollo* and also *toll-6* expression from the salivary gland placode at late stage 10/early stage 11 was controlled through the overall salivary gland morphogenesis program initiated by the upstream homeotic transcription factor Scr, and we wanted to test if within the transcriptional specification cascade downstream of Scr this effect required Fkh, a transcription factor that is key to the early morphogenetic changes [3,4,14]. In *Scr[4]* mutant embryos at mid stage 11 *tol-8o/tollo* expression was present across the salivary gland placode, when in the control in was already specifically excluded from it, indicating that Scr activity is responsible for the expression exclusion as part of the gland morphogenetic program (Fig 6D). By contrast, in *fkh[6]* embryos, *toll-8/tollo* expression exclusion still occurred (S7C Fig), suggesting that other factors downstream of Scr control the ceszation of expression of *toll-8/tollo*. Hairy (h), a transcriptional repressor of the pair-rule gene family [67], and has previously been shown to play a role in salivary gland morphogenesis [68]. In particular, *hairy* mutant glands show invagination and lumen phenotypes highly reminiscent of the Toll-8/Tollo- and Toll-6-overexpression phenotypes described above [68]. We therefore analyzed both *toll-8/tollo* and *toll-6* expression in *hairy* mutant embryos (*h[25]* mutants; Figs 6E and S7D) compared to *fkh* expression. Compared to the control at mid stage 11, where both *toll-8/tollo* (Fig 6E) and *toll-6* (S7D Fig) expression is lost from the region of *fkh* expression, the expression is retained for both in the *h[25]* mutant embryos, strongly suggesting that the Hairy repressor is involved in expression regulation of both *toll-8/tollo* and *toll-6* in the early salivary gland placode.

## Discussion

Complex forms during development arise from simple precursor structures, guided through detailed instructions that are laid down, at least initially, through a transcriptional blueprint and dynamic transcriptional changes. We know that

for the tubulogenesis of the salivary glands in the fly embryo, transcriptional changes are key. Lack of the top-most transcription factors in the hierarchy, Scr, Hth and Exd, leads to complete lack of the glands and their primordia, but more revealing, overexpression of Scr leads to ectopic glands in anterior segments where Scr is not repressed, strongly sugges that the whole morphogenetic cascade can be initiated by patterned expression of one factor alone in early embryogenesis [8,41]. We also know that early morphogenetic macroscopic changes in the tissue are due to a delicate patterning of cell behaviors, including apical constriction and cell intercalation. Their patterning in the tissue primordium involves two key transcription factors [4,5]. What has been lacking thus far was a more complete description of all transcriptional changes occurring in the placode primordium that could help guide our understanding of how other changes and behaviors are implemented that are key to the tubulogenesis and also later function of the tissue.

The single cell expression atlas of the salivary glands at early stages of morphogenesis presented here achieves this. Pseudo-bulk differential expression analysis identified many previously unknown factors whose role in the process will form the basis of future studies. The focus on the early stages of salivary gland tubulogenesis that this single-cell atlas provides adds value to the study of this model process of tube budding, because other approaches at determining the transcriptional changing landscape of *Drosophila* embryos have limitations when it comes to salivary gland placodal cells. With a primordium of only about 200 cells in total within an embryo at stage 10–11 of about 48,000 cells, capturing enough salivary gland placodal cells in a whole embryo approach is always going to be challenging [22–25]. Even if cells were captured, the stages analyzed in whole embryo approaches usually did not span the time window we wanted to capture, i.e., the onset and early moprhogenesis [25]. Hence, our single cell atlas provides a unique insight into this aspect of embryogenesis. We directly compared our dataset to those published by Peng and colleagues [24] and Calderon and colleagues [22] that both captured early salivary gland cells (see Table 1). This comparison revealed that (1) our FACS-enrichment approach allowed us to isolate significantly more salivary gland cells than both other datasets, that (2) our analysis captured many more differentially expressed genes (as defined by both our and also Peng and colleagues's approaches), that (3) we identified many genes previously found to be expressed by in situ (BDGP database) in the salivary glands, but even more that this resource had not identified as salivary gland. Although the total number of genes as a reflection of sequencing depth identified by Peng and colleagues is larger, many of these were only identified in their dataset and are not reflected in the number of differentially expressed genes and therefore might not be salivary gland specific. The high number of differentially expressed genes identified in this study, combined with the larger number of cells and clusters likely representing developmental timelines we feel sets our study apart and makes it a valuable resource for the community.

Genes expressed in salivary gland (normalizing gene names over datasets via FlybaseID validator):

| Genes expressed in salivary gland | This study (0.17 res SG) | Peng and colleagues (SG) | Calderon and colleagues (SG) |
|---|---|---|---|
| Total genes | 659 | 2092 | 392 |
| Shared with this study only | 140 | 348 | 38 |
| Shared with Peng and colleagues only | 347 | 1,546 | 65 |
| Shared with Calderon and colleagues only | 38 | 65 | 156 |
| Shared between all three | 133 | | |

Genes differentially expressed in salivary gland (>0 log2foldchange <0.05 p-value adjusted):

| | This study (0.17 res) | Peng and colleagues. | Calderon and colleagues. |
|---|---|---|---|
| Total genes | 247 | 157 | 70 |
| Shared with May Only | 143 | 67 | 16 |
| Shared with Peng Only | 66 | 48 | 7 |
| Shared with Calderon Only | 16 | 7 | 25 |
| Shared between all three | 22 | | |

**Table 1. Comparison between single cell sequencing datasets identifying early salivary gland cells.**

| | This study (GEO: GSE271294) | | | | | | Peng and colleagues, 2024 (GEO: GSE234602) | Calderon and colleagues, 2022 (GEO: GSE190147, NNv2, pip > 4) |
|---|---|---|---|---|---|---|---|---|
| **Resolution** | N/A | 0.17 | 0.3 | | | | as GEO | as file* |
| **Cluster name** | SrcGFP vs. ArmYFP | Salivary gland | Early progenitor/ epidermal | Speci-fied duct | Specified secretory cells | Post specifica-tion/late SG | Salivary gland manually labeled cluster | Salivary gland |
| Dataset/stage | Stage 10–12 | | | | | | Stage 10–12 | 6–8 hours |
| Number of cells | 3,452 vs. 2,527 | 2,127 | 1,189 | 224 | 1800 | 653 | 114 | 478 |
| Percentage of dataset | 57.7% SrcGFP | 35.5% | 19.8% | 3.7% | 30.1% | 10.9% | 0.55% | 0.62% |
| Number of genes expressed: *Findmarkers +* *(>0 log2fold change in cluster compared to dataset, 10% cells in cluster)* | 642 | 659 | 676 | 955 | 731 | 598 | 2092 | 392 |
| Number of genes differentially expressed, using the limit of Peng and colleagues: *Findmarkers +* *(>0 log2fold change, <0.05 adjusted p-value, gene x expressed in 10% of cells in cluster)* | 346 | 247 | 242 | 77 | 228 | 125 | 157 | 70 |
| Number of genes differentially expressed using the limit of this study: *Findmarkers +* *(>0.25 log2fold change, <10E-25 adjusted p-value)* | 84 | 53 | 62 | 12 | 43 | 9 | 32 | 12 |

*Calderon and colleagues/content/members/DEAP_website/public/RNA/update/seurat_objects/pred_windows/NNv2/6_8_finished_processing.Rds

Genes differentially expressed in salivary gland (>0 log2foldchange <0.05 p-value adjusted) in May and colleagues and the BDGP list of salivary gland expressed genes filtered for "embryonic salivary gland, embryonic salivary gland body, embryonic salivary gland common duct, embryonic salivary gland duct".

| | This study (0.17 res) | BDGP salivary gland expression |
|---|---|---|
| Total genes | 247 | 300 |
| Unique | 166 | 219 |
| Shared | 80 | |

As we anticipated and aimed for, using our pipeline we identified many components that were expressed at specific early stages of gland morphogenesis, and that when analyzed in in situ hybridization showed gland-specific expression either across the primordium (*pip, ogre, pasilla, fili*) or even in spatially restricted patterns within the placode across time (*sano, fog, cv-c*) that we will explore for their significance in the future. Unexpectedly, we also identified a cluster of genes expressed prior to and during specification of the salivary gland placode across parasegment 2 that then all became specifically downregulated in and excluded from the placodal cells fated for secretory function, but not the future duct cells or the surrounding epidermis (*hth, tollo, fj, grh*). This suggested that proteins encoded by these genes might in fact interfere with normal morphogenesis or specification of cells. For Hth, this exclusion of expression following its initial requirement for specification early on had been described [6]. *hth* expression is both regulated by Scr but also together with ExD

upstream of maintaining *Scr* expression in the placode early on. The trio of Scr/Hth/ExD is key for the expression of the next layer of transcription factors [10], including Fkh and Hkb as two factors that are key for initiating part of the early morphogenetic changes [4,14,15].

A second group of genes identified that are also initially expressed in parasegment 2 and then become downregulated and excluded from the secretory cells of the salivary gland placode all belong to the E(spl) cluster of genes encoding transcriptional repressors involved in restricting neurogenic potential downstream of Notch in many tissues [57,58]. From our pseudotime analysis we suggest two pathways of differentiation from a central cluster of 'epithelial and early salivary gland cells' towards either secretory or duct fate. A third possible trajectory could involve some of these epithelial cells differentiating towards neuroectodermal (the E(spl)-cluster; [69]) and then neuronal fate (Fig 4E). Such path of differentiation, usually repeated in a segmental pattern within the embryonic epidermis, might also need to be eliminated from the salivary gland placode so as to not interfere with their morphogenesis, and thus could be the reason for the downregulation and exclusion of expression of E(spl) genes from the secretory cells of the salivary gland placode.

The exclusion of expression of *toll-8/tollo* and *toll-6* is particularly intriguing as Toll/LRRs have been implicated in the regulation of junctional actomyosin accumulation by several studies, including in regulating germband extension in the early *Drosophila* embryonic epidermis [60–62]. Furthermore, Toll-2/18w has been directly implicated in salivary gland morphogenesis, and mutants in *toll-2/18w*, and especially zygotic double mutants in *toll-2/18w* and *rhoGEF2* or *fog* show late gland phenotypes highly reminiscent of those we observed in the case of the continued expression of Toll-8/Tollo in the placode [43]. In the wild-type the gradually expanding expression pattern of Toll-2/18w across the placode is key to wild-type ordered internalization of placodal cells, likely through effects that Tolls/LRRs and therefore Toll-2 can exert on apical actomyosin accumulation and function. Loss of Toll-2/18w perturbs this, as does its overexpression that homogenizes any graded expression that is required for wild-type gland morphogenesis. Thus, the exclusion of *toll-8/tollo* expression could serve to prevent Toll-8/Tollo from interacting and interfering with Toll-2/18w's role, as LRR receptors can undergo homophilic or heterophilic interactions within this family [64]. Toll-8/Tollo, Toll-2/18w and Toll-6 have been shown to be able to physically interact with each other heterophilically between neighboring cells in S2 cell aggregation assays [61]. The fact that continued placodal expression of the TolloΔcyto, i.e., Toll-8/Tollo lacking it cytoplasmic tail, resulted in the same phenotype as overexpression of the full-length version suggests that the intracellular domain is not required to elicit the overexpression effect and that rather an interaction of the extracellular domain might titrate a required factor such as Toll-2/18w from its native interactions, and this could be the reason for the morphogenetic problems. Toll-2/18w is not the only LRR expressed in the placode and involved in salivary gland tubulogenesis. We previously identified Capricious as an LRR protein whose overexpression results in a strong salivary gland tube defect [13]. Using beta-galactosidase P-element traps we concluded that Capricious was endogenously expressed in tissues surrounding the salivary gland cells and Tartan, its usual interacting LRR [70], was expressed in placodal and gland cells itself. *tartan* and *capricious* double mutants show highly aberrant gland lumens [13]. Thus, ectopic LRR expression in the placode could interfere with the endogenous expression and requirement of several LRRs.

What regulates these drastic changes to downregulation and exclusion for genes potentially interfering with the salivary gland morphogenetic program? The exclusion of *toll-8/tollo* expression being downstream of tissue-specifying homeotic factor Scr identifies this as part of this program. Downstream of Scr, a second layer of transcriptional regulation controls morphogenetic effectors and expression of factors related to later secretory function [3,19,34,68]. We show that the downregulation is independent of Fkh, one of two main transcription factors shown to initiate changes to cell and tissue shape during the morphogenesis [4,5,14]. Rather, the downregulation of factors within secretory cells appears controlled by the transcriptional repressor Hairy that was also previously shown to be involved in salivary gland morphogenesis [68], and intriguingly the gland phenotypes of hairy mutant embryos, with aberrantly shaped, too wide and branched lumens, is highly reminiscent of the phenotypes observed when Toll-8/Tollo and Toll-6 are overexpressed.

In summary, our single-cell atlas of early salivary gland tube morphogenesis is a rich source for the identification of dynamically controlled transcriptional changes of known or suspected morphogenetic effectors within the forming salivary glands, and will provide the basis for many further studies in the future.

## Materials and Methods

### Drosophila stocks and husbandry

The following fly stocks were used in this study:

w;;fhkGal4 UAS-srcGFP [13]; Armadillo-YFP (PBac{681.P.FSVS-1}arm^CPTI001198, w^1118;;)(Kyoto Stock Centre/DGGR); Tollo-YFP (BAC) [61]; the following stocks were obtained from the Bloomington Drosophila Stock Centre: E(spl)mγ-HLH-GFP (#BL66401); E(spl)m3-HLH-GFP (#BL66402ß); UAS-TolloFL-GFP (#BL92990); UAS-TolloΔcyto-GFP (#BL92991); UAS-Toll2-FL-EGFP/Cyo (#BL92996); UAS-Toll6-FL-EGFP/CyO (#BL92993); Scr[4] (#BL942; y[1]/Dp(3;Y)Antp[+], y[+]; Scr[W] Scr[4] p[p] e[1]/TM2); fkh[6] (#BL545; y[1] w[*]; fkh[6] mwh[1] Diap1[th-1] st[1] kni[ri-1] rn[roe-1] p[p] cu[1] sr[1] e[s]/ TM3, P{ry[+t7.2]=ftz-lacZ.ry[+]}TM3, Sb[1] ry[*]); h[25] or hry[25] (#BL545; ru[1] hry[25] Diap1[th-1] st[1] cu[1] sr[1] e[s] ca[1]/TM3, Sb[1]).

For expression in the placode, UAS stocks were combined with fkhGal4 that is specifically expressed in the salivary placode and gland throughout development [26].

### Embryo collection pipeline and Flow Cytometry

*Drosophila melanogaster* embryos expressing GFP in only the salivary gland (w;;FkhGal4 UAS-srcGFP) or expressing YFP in all epidermal cells (PBac{681.P.FSVS-1}arm^CPTI001198, w^1,118;;) were collected at 25°C in a humidity and $CO_2$-controlled environment in 17 cages (973 cm$^3$) per genotype. Three one-hour pre-lays were discarded prior to collection to reduce embryo retention in female flies and synchronize egg laying. Embryos were collected in one and two-hour time windows on apple juice agar plates with a small amount of yeast paste. Plates were removed from cages and incubated at 25°C for 5 hours 15 min. Embryos were washed into a basket and incubated in 50% bleach for 3 min for dechorionation and extensively washed. Embryos were removed from the basket and placed on cooled apple juice agar plates to slow developmental progression and visually screened using a fluorescence stereoscope with GFP filter. Embryos displaying an autofluorescent pattern indicating development beyond stage 12 were removed and only younger embryos up to this stage (10–12) were retained.

Mechanical dissociation of stage 10–12 embryos was performed using an adjusted method [23]. Approximately 5,000 embryos were placed in a 1 ml Dounce homogenizer containing 500 µl of ice-cold Schneider's insect medium (Merk). Embryos were dissociated using 8 strokes of a loose pestle, followed by 10 gentle passes through a 16G 2-inch needle (BD microlance 3) into a 5 ml syringe. The final pass was filtered through a 50 µm filter (Sysmex) into a flow cytometry compatible tube. An additional 500 µl of ice-cold Schneider's insect medium was added to the Dounce homogenizer and the process was repeated to retrieve any additional cells to make a total of 1 ml of embryonic single cell suspension. To check single cell suspension was achieved, 20 µl of the suspension was mixed with an equal volume of trypan blue (Thermo), and placed on a CellDrop cell counter (DeNovix) to check live/dead ratios, and to visually assess that single cell suspension had been achieved and no large debris remained. Propidium Iodide (Thermo) was added to a final concentration 10 µg/ml and mixed into the suspension.

Single cells were sorted on a Sony Synergy system with gating for live cells, GFP signal and absence of autofluorescence signal. Live cells were identified by absence of propidium iodide signal, live cells were plotted on a secondary gate to isolate G/YFP+ cells from nonfluorescent cells and debris. This gate was determined by plotting signal from 488 nm laser with a 525nm filter against signal from 488 nm laser with a 510 nm or 405 nm filter. Prior to sorting, *yellow white* embryos were subject to an identical collection dissociation protocol and the boundary between non-GFP and G/YFP+ cells was determined by the maximum detected signal of *yellow white* embryos plotted on the same graph indicating

non-GFP cells. G/YFP+ cells were collected in a 1.5 ml tube containing 37 µl of PBS and placed on ice before single cell sequencing. Three batches of cells were collected, two batches of *srcGFP* cells and one batch of *armYFP* cells. A total of 10,149 cells were collected for ArmYFP_1 and 10,000 for SrcGFP_1 and for SrcGFP_2 6,800 cells were submitted for sequencing to the Cancer Research UK sequencing facility for 10X library preparation and sequencing, according to the manufacturer's protocols.

### Single cell RNA library preparation

Single cell suspensions were processed by Cancer Research UK, Cambridge, for 10X Chromium single cell sequencing. SrcGFP_1 and ArmYFP_1 were run on a singular lane of a NovaSeq flow cell at a 1:1 equimolar ratio with 10X v3.0 technology and sample preparation. SrcGFP_2 was run on a NovaSeq flow cell with two additional samples in an equimolar ratio of 3:1:3, the remaining two samples were excluded due to low cDNA quality. SrcGFP_2 was run on 10X v3.1 technology and sample preparation.

### RNAseq analysis

RAW FastQ files were obtained from the sequencing runs and a modified CellRanger 5.0.1 (10X Genomics) pipeline was applied to generate files for downstream analysis. Reads were aligned to a custom reference genome file generated using CellRanger mkref; this genome consisted of protein coding and antisense genes from the *Drosophila melanogaster* genome build 6.32 obtained from the FlyBase (www.flybase.org) FTP site and two additional custom gene transcripts (GFP and YFP). Firstly, a filtered GTF file was generated using CellRangers mkgtf package for custom reference genome building, the dmel6.32 GTF file from FlyBase, filtered for gene annotations selecting biotypes of protein coding and antisense genes. Following GTF file generation, GFP and YFP transcript sequences were obtained from FlyBase, and added to the newly constructed GTF file. A reference genome file was then built using the custom GTF file and FASTA sequence file for the dmel6.32 genome build. Reads were aligned to the custom genome using CellRanger count. The original CellRanger outputs will be available on GEO.

### Seurat final object generation

All scripts run on CellRanger outputs and can be found in this paper's GitHub repository (https://github.com/roeperlab/SalivaryGland_scRNAseq). All sequencing analysis was carried out using R version 4.2.1 (R Core Team, 2022, https://www.R-project.org/), RStudio Build 576 (R Studio Team, 2020, http://www.rstudio.com/). CellRanger outputs were placed into Seurat objects, using the Seurat v4.1.3 package in R [71]. Briefly, three samples were sequenced and used for subsequent analysis, two batches of salivary gland labeled cells and one batch of epidermal tagged cells. Prior to the merging of datasets, the following limits were applied to all three datasets: cells containing more than 200 but less than 2,500 genes, cells containing less than 100,000 counts and cells containing less than 10% mitochondrial reads were retained in the dataset. Percentage mitochondrial reads were obtained by specifying mitochondrial genes as genes with the prefix "mt:". An additional parameter of ribosomal gene percentage was obtained in a similar manner by specifying ribosomal gene reads to any gene beginning with "RpL" or "RpS". Datasets were combined into a single Seurat object, and the remaining cell reads normalized using ScTransform with integrated anchoring methods, using 3000 genes to generate an anchor list [72]. Resolution of the combined dataset was decided via incrementally increasing resolution during FindAllClusters step and via plotting the splitting of clusters using Clustree v.0.5.0 [73]. For the generation of the total cell dataset, a further investigation into cells present in cluster 0, when including 44 dimensions at a resolution of 0.3, revealed no further clusters of relevance when increasing the resolution value. This cluster is likely a result of high RNA background within the dataset and cells assigned to this cluster were deemed low quality and removed from the dataset. The cells remaining from this pipeline were used for further analysis and a repeat analysis was carried out as above following the removal of low-quality cells, the outcome of this analysis will be referred to as the complete cell dataset. The total dimensions used

to generate the complete cell UMAP was 11 and plotted at a resolution of 0.17 and for the salivary gland development lineage was 0.3.

## Cluster identification

Initial cluster identification on the complete cell dataset was performed by running FindAllMarkers from the Seurat package, with literature reviews carried out for top markers of each cluster. In cases where cluster identity was not clearly assignable based on literature review or via tissue expression annotation in BDGP (Berkeley Drosophila Genome Project https://insitu.fruitfly.org/) or Fly-FISH (https://fly-fish.ccbr.utoronto.ca), in-situ hybridization probes were obtained and the salivary gland region imaged across embryos at varying stages of salivary gland development in order to assign salivary or nonsalivary gland identity.

For the more highly clustered salivary gland lineage, FindAllMarkers was used to generate a new marker gene lists for the newly identified clusters, a literature review was conducted and top markers from clusters were investigated in a similar manner. Top markers from this analysis and their respective $Log_2$Fold change and adjusted p-values from this analysis were also used for DotPlot and VolcanoPlot generation. For additional comparisons between the early cluster and later clusters of the salivary gland lineage, $Log_2$Fold change and adjusted p-values were obtained by running FindMarkers from the Seurat package, with the cluster of interest specified as 'ident.1′ and the early cluster cells as 'ident.2′. Genes of interest *hth, fj, Tollo, grh* and *E(spl)* cluster components were labeled on the resulting Volcano Plots. Volcano plots were generated using the R package EnhancedVolcano v.1.16.0 plotting the output of FindAllMarkers or FindMarkers gene marker tables (Table 2).

## Pseudotime analysis

Pseudotime analysis was carried using a combination of Monocle3 [74], Slingshot [75], and TradeSeq [76] packages. The complete cell dataset was converted into a cell_dataset containing all metadata generated in Seurat including cell embeddings, reductions and cluster information to generate an object compatible with the Monocle package. The dataset was partitioned with a group label size of 3.5 to reflect the UMAP clustering [77]. Using monocles learn_graph function, cells were assigned a pseudotime value and were ordered, root nodes were defined as the nodes covering the epidermal and early salivary gland cell population, prior to any predicted lineage splits. In order to reduce user bias final nodes were not specified. Cells were then plotted as a UMAP coloured by their assigned pseudotime value. For the generation of the pseudotime ordered heat map, cell embeddings from the complete cell dataset were used to generate a Slingshot object, a total of 6 lineages were identified with no starting or final cluster specified, of these 6 lineages, lineage 1 closely matched the predicted salivary gland lineage, all further analysis was carried out using this lineage. From lineage 1 a curve was generated using 150 points and a shrink value of 0.1 with 0 stretch. Using this curve pseudotime and cell weights were generated using Slingshot features slingPseudotime and slingCurveWeight, respectively. These values were then inputted into TradeSeqs NB-GAM model, by running fitGAM with 6 knots. Genes with high differential expression with respect to pseudtime were chosen for heatmap generation, their gene expression averaged and smoothed using predictSmooth from TradeSeq and the resulting data plotted in a heatmap using the pheatmap package (https://CRAN.R-project.org/package=pheatmap) with gene names and cells arranged via their pseudotime assigned value.

## Immunofluorescence analysis

Prior to immunoflourecene labeling embryos were fixed in a 4% formaldehyde solution and stored in 100% methanol at −20°C, or for embryos to be imaged with Rhodamine-phalloidin embryos were stored in 90% ethanol in water at −20°C. Embryos were rehydrated in PBT (PBS, 0.3% TritonX-100) followed by a 5 min incubation in PBS-T (PBS, 0.3% TritonX-100, 0.5% bovine serum albumin) at room temperature. Embryos were blocked in PBS-T for a minimum of 1 hour at 4°C. Primary antibody solution was applied at varying concentrations (see reagent table) and incubated overnight at

**Table 2. Methods and cell groups analyzed for differential expression analyses.**

| Table | Figure | differential expression analysis method, cells included and reasoning | average log2fold change in expression for genes between | |
|---|---|---|---|---|
| | | | group 1 | group 2 |
| S1 Table | Fig 1C. | Cells included in calculation: All cells in the dataset (5,979), labelled by genotype of origin<br>Method: Seurat package **FindAllMarkers**<br>Reasoning: To identify marker genes between cells isolated from salivary gland marked genotype and epithelial tagged genotype. | SrcGFP (Salivary gland) | ArmYFP (Epithelial) |
| S2 Table | None. | Cells included in calculation: All cells in the dataset (5,979), labelled clusters at resolution 0.17<br>Method: Seurat package **FindAllMarkers**<br>Reasoning: To identify marker genes for all clusters in the dataset. Using these genes for literature review to identify cell types within the dataset. | Sequentially, all clusters in dataset | Remaining cells in the dataset which are not in the cluster. |
| S3 Table | Fig 3B and 3C | Cells included in calculation: All cells in the dataset (5,979), clustered and labelled at resolution at 0.3<br>Method: Seurat package **FindAllMarkers** – generated markers for all clusters in the dataset. Table outputs information for all 10 clusters in the dataset, Fig 3B and 3C plots values for genes upregulated/downregulated for the four clusters identified in the salivary gland linage: (early, duct, secretory, specification)<br>Reasoning: To identify marker genes for each cluster in the suspected salivary gland lineage of cells. | Early cluster | Remaining cells in the dataset |
| | | | Specified duct | Remaining cells in the dataset |
| | | | Specified Secretory | Remaining cells in the dataset |
| | | | Post specification | Remaining cells in the dataset |
| S4 Table | Fig 3F | Cells included in calculation: All cells in the dataset (5,979)<br>Method: monocle, tradeseq, slingshot packages – lineages generated using learn_graph – 6 nodes in early nodes chosen as first nodes.:ineage 1 chosen as passes through SG clusters and pseudotime values for each cell with slingshot, differentially expressed genes across lineage also generated with slingshot<br>Reasoning: Identify genes with variable expression across the linenage running through the salivary gland related clusters | Every individual cell | Every other cell in the dataset |
| S6 Table | S3B Fig | Cells included in calculation: Cells within the epidermal and early salivary gland cells Cluster at resolution 0.17 only (1,663 cells)<br>Method: Seurat package **FindAllMarkers** – generated markers for all clusters in the dataset. Markers and Log2fold change for the newly emerged duct cluster included in S3B Fig.<br>Reasoning: To identify the marker genes of a new cluster that occurring in these cells with increased clustering to resolution 0.3. | New cluster (duct) | Remaining cells in the epidermal and early salivary gland cells |
| S6 Table | S3C Fig | Cells included in calculation: Cells within the epidermal and early salivary gland cells Cluster at resolution 0.17 (1,663 cells)<br>Method: Seurat package **FindMarkers** – Comparison between cluster X and cluster Y.<br>To identify the marker genes of new clusters occurring in these cells with increased clustering. Compared cluster X to cluster Y, because two new clustered had appeared and we wanted to specifically know what differentiates these two clusters from each other rather than genes are different compared to all cells in the dataset. | Cluster X | Cluster Y |
| S5 Table | Fig 4A. | Cells included in calculation: Cells clustered at resolution 0.3 (Fig 3A) (5,979 cells)Method used: Seurat package **FindMarkers** – ident.1 is cells in column group 1 and ident.2 cells listed in group 2 column.<br>Reasoning: Fig 3B and 3C look at markers in clusters when comparing one cluster to the entire dataset. In this figure we wanted to track which genes are upregulated and downregulated compared to the earliest salivary gland fated cells, rather than comparing to the entire dataset as a whole.<br>Fig 4A early uses data for early compared to entire dataset to show the genes which are expressed in the earliest salivary gland. Four of which are labelled. The remaining 3 plots in Fig 3A shows genes which are upregulated when comparing a specific cluster in the SG lineage to the early salivary gland cells. To show how the expression of the original early top marker genes changes over the progression of time, the four genes are labelled in each comparison volcano plot. | Early cluster | Remaining cells in the dataset |
| | | | Specified duct | Early cluster |
| | | | Specified Secretory | Early cluster |
| | | | Post specification | Early cluster |
| S2 Table (e(spl) sheet). | Fig 4C | Groups labelled as: Cells clustered at resolution 0.17 (Fig 2A)<br>Method used: Seurat package **FindAllMarkers** – filtered the output table for each cluster that is relevant (E(spl) cluster<br>To identify genes significantly upregulated or downregulated within the cluster compared to all other cells in the dataset. | E(spl)cluster | Remaining cells in the dataset |

4°C. The following morning primary antibody solution was removed and two washes in PBS-T were carried out at room temperature followed but 3 longer 20 min washes before secondary antibody solution was applied and incubated for 1.5–3 hours. For immunofluorecene labeling containing Rhodamine-phalloidin, phalloidin was included in the secondary antibody solution. The secondary antibody solution was removed and a further two washes in PBS-T were carried out followed by three longer 20 min washes before a final wash in PBS for five minutes. Embryos were mounted in Vectorshield (Vectorlabs H-1000) before being imaged. All immunofluorescence images were captured on an Olympus FluoView 1,200 confocal microscope using a 40x oil objective.

## HCR analysis

Probes and hairpins for HCR in-situ hybridization for genes of interest were obtained from Molecular Instruments, NM accession numbers were specified and in cases where genes had multiple isoforms regions of transcript shared amongst all transcripts were used to request probes. Embryos were collected and fixed in 4% Formaldehyde as detailed above, and stored in 100% methanol at −20°C before following a whole mount embryo HCR protocol [78]. Batches of embryos were pooled and rehydrated in PBS + 0.1% Tween-20 (PBS-TW). Embryos were pre-hybridized at 37°C in hybridization buffer followed by an overnight incubation at 37°C in primary probe solution (probe diluted to 0.8 µM in hybridization buffer). The following morning the probe solution was removed and embryos washed in probe wash buffer four times in 15 min increments at 37°C followed by two short 5 min washes in 5x SSCT buffer at room temperature. Secondary probes were chosen based on primary probe amplification region and the secondary probes' emission signal. Care was taken to move any salivary gland probe markers to 647nm as to not overly saturate channels during imaging. Secondary probes were treated as individual hairpins (H1 and H2) initially and were separately heated to 97°C for 90 seconds before snap-cooling to room temperature. Embryos were amplified in room temperature amplification buffer for 10 min before combining H1 and H2 in amplification buffer to a final concentration of 0.8 µM before being added to the embryos and incubated overnight in the dark at room temperature. The following morning secondary probes were removed and embryos washed in 5xSSCT for a 5 min wash followed by two 30 min washes and a final 5 min wash. All buffer was removed and embryos were mounted in VectaShield mounting medium containing DAPI (Vectorlabs H-1000) for immediate imaging. All in situ hybridization images were captured as z-stacks on a Zeiss 710 Upright Confocal Scanning microscope with a 40x oil objective using full spectral imaging, and images were post-acquisition linear un-mixed. For linear unmixing using the Zeiss software individual spectra for each probe wavelength were obtained by carrying out in-situ hybridization for highly expressing salivary gland genes (*fkh, CrebA*), one gene was chosen and one probe wavelength chosen. Regions of high signal were then specified and used to obtain spectral readings for the respective wavelength (488, 594nm or 647nm). *ArmYFP* embryos were scanned unstained to obtain the YFP spectrum and for the DAPI spectrum nuclei from *white* embryos mounted in VectaShield containing DAPI were used.

For early salivary gland development stages 10 and early stage 11 (pre-apical constriction, apical constriction) HCR was carried out on ArmYFP embryos in order to simultaneously image membrane labeling alongside mRNA expression, for these cases secondary probes at 488nm were not used.

## Quantification of apical area

For the analysis of apical cell area, images of fixed embryos of the genotypes w;;*fkh-gal4* embryos and w;;*fkh.gal4, UAS-TolloGFP* labelled with PY20 and phalloidin-Alexa633 were analyzed. Images were categorized into apical constriction and continued invagination based on total cell numbers at the surface of the placode and surrounding morphological features. The first 4 optical sections (covering 4µm in depth) to display PY20 membrane signal were compiled into a maximum intensity projection and the PY20 membrane signal used for segmentation of the apical cell boundary. Segmentation was carried out with the Fiji plugin TissueMiner "*Detect bonds V3 watershed segmentation of cells*" [79]. The following parameters were used during segmentation: despeckling/strong blur value of 3/ weak blur value of 0.3/ cells smaller than

5px excluded/ a merge basins criteria of 0.2/kernel diameters in comparison to kuwahara pass and max pass of 5px and 3px, respectively. Automatic bond detection was hand-corrected to remove over-segmented cells and introduce missed bonds. Cells determined to be outside the placode were excluded. Cell data were exported from TissueMiner and reported cell areas were converted from px to $\mu m^2$. Frequency distribution of cell areas were calculate and plotted using GraphPad Prism (version 9.5.1 for MacOS, GraphPad Software, Boston, Massachusetts USA). A histogram of the cell areas with bins of $2\mu m^2$ was generated and the values plotted as a cumulative percentage of cells.

**Quantification of apical F-actin.** For the analysis of apical F-actin, images of fixed embryos of the genotypes w;;*fkhGal4*, w;;*fkhGal4; UAS-TolloGFP* were stained with PY20 to label apical cell outlines and Rhodamine-phalloidin to label F-actin. To assess the difference in phalloidin signal (F-actin) across the apical placodal area, for each analyzed placode/embryo, using projections of the apical-most 4 confocal sections (each of 1µm thickness), three 10µm x 10µm areas both inside and outside the placode were quantified for fluorescence intensity per area, averaged and expressed as an inside/outside-ratio that was plotted.

**Quantification of *toll-8/tollo* HCR signal.** For the analysis of *toll-8/tollo* HCR signal, images of fixed wild-type embryos were analyzed processed for HCR in situ for *toll-8/tollo* and *fkh*. To be able to identify differences in fluorescence signal intensity of the probes between different stages, the ratio in fluorescence intensity per area of *toll-8/tollo* HCR of a similarly sized ROI between parasegment 2, where the salivary gland placode is localized as identified by *fkh* expression, and parasegment 1, where *toll-8/tollo* expression does not vary across the time period, was determined.

Projections of the confocal sections covering the placodal cells were used.

## Statistical analysis

For comparison of cumulative distributions in the analysis of apical area size a Wilcoxon matched-pairs signed rank test was used (as the distribution of apical area values is assessed for two conditions, but normality of the distributions cannot be assumed), the relevant p value is indicated in the figure legend. For analysis of apical F-actin enrichment compared between different genotypes, differences in ratio values for fluorescence intensity inside/outside placode were tested for significance using un-paired Student *t* test (to compare the mean between the two independent groups/genotypes), mean +/- SEM are plotted and p-value is indicated in the figure legend. For analysis *toll-8/tollo* HCR fluorescence intensity per area ratio of parasegment 1/parasegment 2 compared between different embryonic stages (late stage 10 versus mid stage 11), differences in ratio values for fluorescence intensity were tested for significance using un-paired Student's *t* test (to compare the mean between the two independent groups/embryonic stages), mean +/- SEM are plotted and p-value is indicated in the figure legend.

## Supporting information

**S1 Fig. Related to Fig 1.** Generation of a single cell transcriptome dataset of salivary gland placodal and epidermal cells. **A a, b)** At mid stage 10, endogenous Fkh protein, revealed using an antibody against Fkh (red), is already spreading across the placode from the initial expression at the forming pit position (asterisks in all panels mark the position of the future pit). At this stage, srcGFP (green) expressed under fkhGal4 control can be clearly identified to begin to be expressed in central cells of the placode at varying levels. **c-d')** At late stage 10 and early stage 11, when endogenous Fkh is seen in all secretory placodal cells (but constriction near the forming invagination point is only just beginning; see cross section panels in **c'** and **d'**), srcGFP expression driven by *fkhGal4* is very strong in nearly all placodal cells, with only slightly lower levels in the most anterior cells. Dotted lines mark the boundary of the placode, arrows and green brackets indicate cells outside the placode that express srcGFP when driven by fkhGal4. Cell outlines are marked by an antibody against junctional phosphotyrosine (PY20; blue). **B)** FACS plots for nonfluorescent control (*w; +;fkhGal4*), two srcGFP embryo batches (srcGFP_1 and srcGFP_2; *w;fkhGal4 UAS-srcGFP*) and one ArmYFP embryo batch (*arm[CPTI001198], w[118]; +;+*) used for single cell RNA-sequencing. The top row shows the sorting for live versus dead cells, the bottom row

shows the gate for sorting of GFP/YFP-positive cells and the percentage of total cells sorted they comprised. **C)** Single channels of the HCR *in situs* in Fig 1F for the indicated genes.
(TIF)

**S2 Fig. Related to** Fig 1**.** Generation of a single cell transcriptome dataset of salivary gland placodal and epidermal cells. **A)** Quality control of scRNA-sequencing batches, showing cut offs (red lines) for number of genes, number of reads, % of reads originating from mitochondrial genes and % of reads originating from ribosomal genes. **B-B')** UMAP of single cell RNA sequencing at cluster resolution 0 (**B**) and 0.3 (**B'**). **B")** Illustrates the emergence and linkage of clusters with increasing resolution generated using the Clustree package in R. Red dotted outline shows clusters assigned at resolutions of 0.2 and 0.3 where further investigations into cluster makers occurred. **C)** General consensus of markers represented in the clusters emerging at resolution 0.3, purple arrows represent the emergency of cell clusters with markers of low-quality cells and blue arrows represent the emergence of clusters with biologically relevant cell types. **C')** Biological process Gene Ontology terms for gene lists generated from resolution 0.3 clusters featured in **B**, ranked by the $-Log_{10}$P-value provided by FlyMine curated lists for each ontology term. **D)** UMAP generated following the reclustering of the original dataset following the exclusion of low-quality cell types identified in **C**. **E)** UMAPs displaying the distribution of number of genes per cell, number of reads per cell, % of reads originating from mitochondrial genes and % of reads originating from ribosomal genes to illustrate that these do not cluster in any particular way across the UMAP.
(TIF)

**S3 Fig. Related to** Fig 2**. Generation of a single cell transcriptome dataset of salivary gland placodal and epidermal cells.** Further analysis of the 'epidermal and early salivary gland cells' cluster identified in Fig 2. **A)** Isolation of the cluster from the resolution 0.17 dataset. The cluster is then further splitting by increasing resolution as illustrated in the flow scheme. **B)** At resolution 0.3 the cluster splits into three, with 244 cells identified as salivary gland, 244 that form a new cluster, and 1189 that remain in the epithelial cluster. Marker genes and published *in situs* (BDGP) for the new (red) cluster suggest this to be a duct cluster, as also further analyzed in Fig 3. **B')** UMAPs of selected top marker genes for epithelial clusters and new duct cluster identified in **B**. **C)** Reclustering at resolution 0.5 splits the epithelial cluster remaining at resolution 0.3 into two further clusters X and Y, and the listed marker genes and published *in situs* (BDGP) appear to suggest that these clusters could represent more anterior and more posterior epidermis. **C')** UMAPs of top marker genes for clusters X and Y identified in **C**.
(TIF)

**S4 Fig. Related to** Fig 3**.** A single cell timeline of mRNA expression changes during salivary gland morphogenesis. **A-E)** In situ hybridization by HCR of one top marker gene per cluster identified in comparison to *fkh* expression as shown in Fig 3D, single channels are shown here: *hth* for the 'progenitor and early gland' cluster, *CG45,263* for the 'specified duct cells' cluster, *Gmap* for the 'specified secretory cells' cluster and *Calr* for the 'post specification/late' cluster. Single in situ channels matching the panels in Fig 3D are shown, with matching schematics explaining the labeling at continued invagination/ stage 12. White brackets indicate the position of the salivary gland placodes, scale bars are 30µm. **F)** UMAP plots based on the clustering in Fig 2A showing expression of *fkh* and *GFP* in the combined datasets. **G)** Pseudotime analysis based on the proposed salivary gland portion of the lower resolution UMAP in Fig 3A. The pseudotime also agrees with a second shorter lineage progression to ductal fate as one possible trajectory.
(TIF)

**S5 Fig. Related to** Fig 4**.** Placode-specific downregulation and exclusion of expression of candidates. **A)** UMAP plots based on the clustering in Fig 2A showing expression of *hth*, *grh*, *toll-8/tollo* and *fj* in the combined datasets. **B)** UMAP plots based on the clustering in Fig 2A showing expression of *E(spl)mγ-HLH*, *BobA* and *E(spl)m4-BFM* in the combined datasets, note the strong increase in expression in the E(sol) cluster compared to the early epidermal cluster. **C)**

Comparison of expression of *E(spl)m4-BFM*, exemplary for the E(spl) group, between early stage 10 (gland specification) and mid stage 11 (apical constriction), to illustrate the expression in parasegment 2 at the specification stage (*fkh* expression just initiating) and following downregulation and exclusion of expression in the secretory cells once morphogenesis commences. HCR in situ for *E(spl)m4-BFM* is shown in green and for *fkh* in magenta in the overlay, scale bar is 20μm, white brackets indicate the position of the placode. **D)** Direct comparison *of toll-8/tollo* in situ by HCR between gland specification stage (late stage 10) and continued invagination stage (stage12) using inverted black and white panels (panels previously shown in Fig 4F), with the position of the secretory cells in parasegment 2 outlined in magenta for one of the two placodes in each panel.
(TIF)

**S6 Fig. Related to** Fig 5. Continued expression of Tollo/Toll-8 disrupts salivary gland tubulogenesis. **A, B, C)** Schematics of Toll-8/Tollo lacking the intracellular cytoplasmic domain (Δcyto; **A**), of Toll-2/18w (**B**) and Toll-6 (**C**) used for re-expression of in the salivary gland placode using the UAS/Gal4 system. **A', B', C')** In contrast to control placodes (Fig 5B) where apical constriction begins in the dorsal posterior corner and a narrow lumen single tube invaginates from stage 11 onwards in embryos continuously expressing *UAS-TolloΔcyto-GFP* or *UAS-Toll-2/18w-FL-EGFP* or *UAS-Toll-6-FL-EGFP* under *fkhGal4* control multiple initial invagination sites and lumens form and early invaginated tubes show too wide lumens (magenta arrows in cross-section views). Fully invaginated glands at stage 15 show highly aberrant lumens. Apical membrane are labelled with an antibody against phosphotyrosine (PY20) labeling apical junctions. Dotted lines mark the boundary of the placode, asterisks the wild-type invagination point. Green panels show the expression domain of TolloΔcyto-GFP, Toll-2/18w-FL-EGFP and Toll-6-FL-EGFP.
(TIF)

**S7 Fig. Related to** Fig 6. Continued expression of Toll-8/Tollo disrupts an endogenous LRR code required for proper morphogenesis. **A)** At early stage 10, at the very onset of specification of the salivary gland primordium, *toll-2/18w*, *toll-6* and *toll-8/tollo* are still expressed in complementary stripe patterns across the epidermis and all overlap with *fkh* expression. Top row shows in situ hybridizations by HCR for *toll-2*, *toll-6* and *toll-8,* in comparison to *fkh* in situ in the lower panels. Magneta lines indicate positions of parasegmental boundaries, thereby illustrating the overalap of all three *toll* in situ signals with *fkh* expression at this point. **B)** Overexpression of Toll-8/Tollo-FL-GFP under *fkhGal4* control does not affect either *toll-2/18w* or *toll-6* mRNA levels or localization in the secretory cells of the salivary gland placode. Dotted lines mark the position of the secretory cells for one of the two placodes shown. HCR in situ for *toll-2/18w* is in magenta, for *toll-6* is in yellow, and either the overexpressed Toll-8/Tollo-GFP or ArmYFP to identify the placode position are shown in turquoise; scale bars are 20μm. **C)** Comparison of *toll-8/tolo8* expression analyzed by in situ (HCR) in *fkh[6]* mutant embryos and control embryos at stage 11. *Toll-8/tollo* is in green and ArmYFP in magenta. White brackets indicate the position of the placode in parasegment 2, scale bars are 30μm.
(TIF)

**S1 Table. Related to** Fig 1C. List of marker genes identified for cells of salivary gland placodal origin and epidermal origin and accompanying information. Marker genes generated by comparing cells originating from *fkhGal4 x UAS-SrcGFP* embryos sorted for GFP signal (salivary gland placode) and cells originating from *ArmYFP* embryos sorted for YFP signal (epidermis/epithelium) scRNAseq data, as plotted in Fig 1C. Limits applied to dataset for literature review categorization were p-value adjusted ≤ 10E-25 and Log2Fold change ≥ 0.25 or ≤ −0.25. Genes categorized by highest piece or relevant information in the following order: 1) mutant phenotype in the salivary gland (MP), 2) microarray data showing expression or altered expression in the salivary gland in mutant phenotypes (MA), 3) in situ database images displaying expression in the salivary gland (I) or 4) no available information on expression in the salivary gland (New).
(XLSX)

**S2 Table. Related to** Fig 2A**.** Marker genes for the cell type clusters of combined salivary gland and epithelial cells. Marker genes for each individual cluster of the combined dataset at a resolution of 0.17 (see Fig 2A), generated using scRNAseq data. Limits applied to dataset: p-value adjusted ≤ 10E-25 and Log2Fold change ≥ 0.25 or ≤ −0.25. Each cell type cluster is displayed on an individual sheet, additionally data from the E(spl)-enriched sheet is plotted in Fig 4C.
(XLSX)

**S3 Table. Related to** Fig 3A**.** Marker genes defining clusters within the salivary gland temporal lineage. Marker genes for each individual subcluster of the salivary gland lineage at an increased resolution of 0.3 (see Fig 3A), generated using scRNAseq data. Limits applied to dataset: Log2Fold change ≥ 0.25 or ≤ −0.25. Each cell type cluster is displayed on an individual sheet.
(XLSX)

**S4 Table. Related to** Fig 3F**. Differentially expressed genes across pseudotime in the salivary gland lineage.** List of genes which are differentially expressed according to their wald statistic value (waldstat_1), p-value and mean log2fold change across the salivary gland lineage (see Fig 3F) generated from scRNAseq data. Limit: p-value < 0.05.
(XLSX)

**S5 Table. Related to** Fig 4A**.** Marker genes for clusters within the salivary gland temporal lineage when compared to the earliest cluster. Marker genes for each individual subcluster of the salivary gland lineage at a resolution of 0.3 when compared to the earliest cluster of cells (see Fig 4C), generated using scRNAseq data. Limits applied to dataset: p-value adjusted ≤ 10E-25 and Log2Fold change ≥ 0.25 or ≤ −0.25. Each cell type comparison is split into an individual sheet.
(XLSX)

**S6 Table. Related to S3B and** S3C Fig**.** Marker genes for clusters within the epidermal/early salivary gland cluster at resolution 0.17. Marker genes for each individual subcluster of the epidermal/early salivary gland cluster at a resolution of 0.17, generated using scRNAseq data. Limits applied to dataset: Log2Fold change ≥ 0.25 or ≤ −0.25. Each resolution is displayed on an additional sheet.
(XLSX)

**S1 Data. Related to** Fig 4F**'. Raw ImageJ quantification data of fluorescence intensity of toll-8/tollo HCR.**
(XLSX)

**S2 Data. Related to** Fig 4F**'. GraphPad Prism 9 file of fluorescence intensity ratio data derived from** S1 Data **for statistical analysis and plotting (the Prism file version allows the display of statistics and plotting used).**
(PZFX)

**S3 Data. Related to** Fig 5D**'. Raw data of apical area quantification under *fkhGal4* control and *fkhGal4 x UAS-Toll-8/Tollo-GFP*.**
(XLSX)

**S4 Data. Related to** Fig 5D**'. GraphPad Prism 9 file of cumulative analysis of apical area quantification data from 'S3_data.xls' in size bins and statistical analysis (the Prism file version allows the display of statistics and plotting used).**
(PZFX)

**S5 Data. Related to** Fig 5E**'. Raw ImageJ quantification of fluorescence intensity of F-actin in placodal cells.**
(XLSX)

**S6 Data. Related to** Fig 5E'**. GraphPad Prism 9 file of fluorescence intensity ratio data derived from** S5 Data **for statistical analysis and plotting (the Prism file version allows the display of statistics and plotting used).** (PZFX)

## Acknowledgments

The authors would like to thank the following people; for reagents and fly stocks: Debbie Andrew, Jen Zallen, Thomas Lecuit, Sarah Bray. Stocks obtained from the Bloomington Drosophila Stock Center (NIH P40OD018537) were used in this study and we thank them for their efforts.

For the purpose of open access, the MRC Laboratory of Molecular Biology has applied a CC BY public copyright licence to any Author Accepted Manuscript version arising.

## Author contributions

**Conceptualization:** Katja Röper.

**Data curation:** Annabel May, Katja Röper.

**Formal analysis:** Annabel May, Katja Röper.

**Funding acquisition:** Katja Röper.

**Investigation:** Annabel May, Katja Röper.

**Methodology:** Annabel May.

**Project administration:** Katja Röper.

**Visualization:** Katja Röper.

**Writing – original draft:** Annabel May, Katja Röper.

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
