## [Editor Report · Decision Letter 0]

17 Jun 2024

Dear Dr Röper,

Thank you for submitting your manuscript entitled "Single cell transcriptomics of the Drosophila embryonic salivary gland reveals not only induction but also exclusion of expression as key morphogenetic control steps" for consideration as a Research Article by PLOS Biology and please accept my apologies for the delay in sending you an initial decision. I had wished to discuss your manuscript with an Academic Editor who works in this field, and it took a bit longer than normal to find someone who was available to provide advice.

Your manuscript has now been evaluated by the PLOS Biology editorial staff as well as by an academic editor with relevant expertise and I am writing to let you know that we would like to send your submission out for external peer review.

Once your full submission is complete, your paper will undergo a series of checks in preparation for peer review. After your manuscript has passed the checks it will be sent out for review. To provide the metadata for your submission, please Login to Editorial Manager (https://www.editorialmanager.com/pbiology) within two working days, i.e. by Jun 19 2024 11:59PM.

Kind regards,

Luke

Lucas Smith, Ph.D.

Senior Editor

PLOS Biology

lsmith@plos.org

---

## [Decision Letter · Decision Letter 1]

7 Aug 2024

Dear Dr Röper,

Thank you for your patience while your manuscript "Single cell transcriptomics of the Drosophila embryonic salivary gland reveals not only induction but also exclusion of expression as key morphogenetic control steps" was peer-reviewed at PLOS Biology. Your manuscript has been evaluated by the PLOS Biology editors, an Academic Editor with relevant expertise, and by several independent reviewers.

As you will see in the reviewer reports, which can be found at the end of this email, although the reviewers find the work potentially interesting, they have also raised a substantial number of important concerns. Based on their specific comments and following discussion with the Academic Editor, it is clear that a substantial amount of work would be required to meet the criteria for publication in PLOS Biology. However, given our and the reviewer interest in your study, we would be open to inviting a comprehensive revision of the study that thoroughly addresses all the reviewers' comments. Given the extent of revision that would be needed, we cannot make a decision about publication until we have seen the revised manuscript and your response to the reviewers' comments. Your revised manuscript would need to be seen by the reviewers again, but please note that we would not engage them unless their main concerns have been addressed.

Specifically, the reviewers are generally in agreement that your study provides a valuable transcriptomic dataset on the regulation of salivary gland morphogenesis. However, the reviewers raised overlapping concerns with the overall strength of the functional insights provided by the manuscript. After discussions with the Academic Editor, we do agree with the reviewers on this point and we ask that you include further functional validation of the newly identified genes to deepen the analyses. To this end, we suggest that you test some of the hypotheses put forward in the discussion, such as Tollo exclusion and interactions with the LRR receptors. In addition, we appreciate that the reviewers raise concerns with the conceptual novelty of the manuscript given the recent paper published in Development (Peng et al, 2024, PMID 38174902), but please note that any novelty concerns did not factor into our editorial assessment given our scooping protection policy.

We appreciate that these requests represent a great deal of extra work, and we are willing to relax our standard revision time to allow you 6 months to revise your study. Please email us (plosbiology@plos.org) if you have any questions or concerns, or envision needing a (short) extension.

**IMPORTANT - SUBMITTING YOUR REVISION**

*Resubmission Checklist*

*Published Peer Review*

*PLOS Data Policy*

*Blot and Gel Data Policy*

Sincerely,

Suzanne

Suzanne De Bruijn, PhD,

Associate Editor

PLOS Biology

sbruijn@plos.org

REVIEWS:

Reviewer #1: In this work single-cell RNA sequence analysis is used to explore the morphogenetic events that lead to the formation of salivary glands (SG) in Drosophila, which represent an excellent model system for organ formation. Previous work, including work from the Röper lab, have identified many of the cellular and genetic mechanisms of SG morphogenesis. Here the authors establish an improved protocol to isolate SG cells in a selected temporal window and compare them to otherwise epidermal cells. This approach provides them with enriched populations of selected cells for further analysis. This is an important point as the SG population represents a small proportion of the total amount of cells in the embryo. In fact, some previous sc-RNAseq approaches have failed to clearly identify SG cells. Nevertheless, a recently published study has also succeeded in identifying SG cell types (Peng et al, Dev 2024, doi:10.1242/dev.202097). The authors first provide a comprehensive description of upregulated genes, particularly in comparison with the rest of the epidermis. Then they evaluate the dynamic progression of expression in the different clusters identified among the SG population, which revealed patterns of activation and cessation of expression that indicated a fine-tuned temporal regulation. HCR-in situ hybridization is used to confirm several identified genes. Finally, the authors identify a group of genes that become excluded from the secretory placode as SG development proceeds in spite of the initial presence there. They focus on the exclusion of Tollo to investigate the functional requirement of this downregulation. They find that the continued presence of Tollo leads to unregulated apical constriction and invagination defects that correlate with aberrant actomyosin accumulation. They propose that Toll exclusion is necessary to avoid interfering with other LRR receptors.

This work represents a technical advance, providing extremely valuable data and information about the transcriptional blueprint of SG specification. The protocol established, sorting the cells of interest, can be applied to the analysis of different organs and tissues to provide enriched populations for further analysis. However, from the conceptual point of view the advance that this work is providing is much more limited. On the one hand, several of the upregulated identified genes have been already identified, using other classical approaches or by comparable analyses (Peng et al, Dev 2024, doi:10.1242/dev.202097). No functional validation of newly identified factors is presented in this work. On the other hand, the authors highlight the exclusion of specific factors during SG specification. It seems not unusual that besides the upregulation of specific factors, morphogenesis and organogenesis is also linked to the silencing of other factors. Several examples can be found in the literature (e.g. serpent expression in the endoderm disappears at st 10-11, Rehorn et al 1996; doi.org/10.1242/dev.122.12.4023), and also in the SGs themselves, where it is known that fkh is inhibited in the duct cells (Haberman et al 2003, doi: 10.1016/s0012-1606(03)00140-4) or that ribbon transcriptionally activates and represses different factors (Loganathan et al, 2016; doi:10.1016/j.ydbio.2015.10.016). Therefore, selective exclusion of factors during organ formation does not seem to convey a novel or surprising concept. At this stage, this work results preliminary and descriptive, as newly identified genes are not functionally tested, and because the analysis of LRR receptors needs to be further investigated (see below).

Specific comments

1) A recent paper analyzed the organogenetic transcriptomes of Drosophila embryos at different developmental stages (one of them corresponding to the stages analyzed in this work) by sc-RNA seq, with a particular focus on SG. In this published work whole embryos were used as the sample. It would be interesting to compare the results that the authors obtained with those published, to evaluate the advantages of cell sorting and the reproducibility of results upon different experimental treatments.

2) Expression of CG 45263 in duct cells in Figs 3 and S3 is unclear. The authors should point to the expression pattern or use magnifications. It is also unclear whether the fkh-Gal4 used to isolate cells is expressed in the duct primordium.

3) The observations with LRR receptors are very interesting. A more detailed analysis would be needed to strengthen the conclusions drawn. Actually, it would be very interesting if the authors could test some of the speculations that they put forward in the discussion, as several tools seem to be available.

-It is proposed that Tollo is excluded from the SG placode to prevent undesirable interactions with other LRR receptors (possibly through heterophilic interactions). They find accumulation of 18w at the invagination point, suggesting a role of this receptor in the process. Is 18w required for invagination? Does it modulate actomyosin accumulation in the placode?

-The results presented are based on the expression pattern of three Toll receptors (18w, Toll-6 and Tollo), and the phenotype of Tollo overexpression. However, it is not addressed whether the defects of overexpression are due to a specific effect of Tollo or just unspecific effects due to the high overexpression.

In this line it would be interesting to see whether the overexpression of the two other Toll receptors produce similar effects.

-It would be interesting to understand whether Toll receptors are required and interchangeable for SG development. Thus, could Tollo or Toll-6 (also excluded from the SG placode) rescue the defects of 18w?

-It is unclear in the results section the logics of using the Tollo∆cyto construct.

Reviewer #2:

In the study by May & Röper, the authors use single-cell RNA-sequencing (scRNA-seq) to characterize gene expression important for salivary gland development. The salivary gland develops from simple epithelial placodes that are specified by homeotic factors Scr/Hth/Exd. To obtain sufficient quantities of salivary gland precursor cells, the authors use a cell sorting approach based on the expression of GFP in these cells driven by fkh-Gal4 using the GAL4/UAS system. In addition to salivary gland cells (orange cluster Fig 2A), this sorting approach also enriched for other cell types including CNS, muscle, hemocyte, anterior midgut, and non-salivary gland epidermal cells (i.e. other clusters, Figure 2A). Using a combination of hybridization chain reaction (HCR) probes and in situ, the authors assay representative marker genes (i) to provide supporting evidence for identity of cells within clusters and (ii) to make a case for dynamic gene expression associated with salivary gland development. The interpretation of subclusters in terms of dynamics is a reasonable idea, and partially supported by their trajectory analysis. However, the reviewer is concerned that the authors are likely oversimplifying. As cell sorting/fkh-Gal4 likely identified additional cell type, beside the placode cells, how sure are the authors that cells used for the trajectory analysis relate to salivary gland development. In particular, the "early" salivary gland precursor cells (as defined in Fig 3) were first characterized as "non-salivary gland epithelial cells" (Fig 2).

Attention should be made to clarify where and when the fkh-Gal4 driver supports expression, as it is likely expressed in a broader area (perhaps early at least) and perdurance of the GFP could mean that cells besides salivary gland precursors were isolated by FACS and present with the dataset. They recognize this is the case, as CNS, muscle and hemocyte cells were found. However, the relationship of the large clusters - salivary gland, non-salivary gland, and enhancer of split (Fig 2A, orange, light blue, green) should be more closely dissected. If fkh-Gal4 driver is as localized to the salivary gland as shown in Fig 1A, it is unclear how cells other than salivary secretory cells were identified; nevertheless, they were. Is the fkh-Gal4>GFP nonsensitive, or could other cells have arisen in the dataset due to cell doublets? Furthermore, it should be determined whether fkh-Gal4 is able to identify cells as the authors suggest (i.e. stage 10 doesn't show staining in Fig 1A).

In summary, this is an interesting study that showcases the utility of scRNA-seq to characterize gene expression in low-abundance cell types such as salivary gland cells (representing 0.4% of the total). While the authors recognize that their sample is temporally diverse (stages 10-12), they need to examine whether the different clusters identified (Fig 2A) relate to time or space/cell types.

In particular, the authors should explore the possibility that the "non-salivary gland epidermal cells" (Fig 2A) are progenitors also for the enhancer of split enriched cells. If that is the case, Tollo may simply be associated with the designation of two distinct cell types (retained in Espl cell cluster but lost in salivary gland). The message may stay the same, but it seems like an oversimplification as presented to focus on the exclusion of Tollo as an important finding.

The text is clear and the approach is logical. Below are points/questions that if addressed could help to make this interesting study stronger.

Major comments:

1) The focus on Tollo exclusion does not seem well supported. The data shown in Fig 4E is not convincing; and while the results shown in Fig 6A,B are better - this effect needs to be quantified and examined more closely. For instance, if tollo/toll-8 is not excluded from the salivary gland placode then is toll-2 expression affected? Is the toll-2 expression lost? In Fig 6B, tollo expression appears to be retained at the bottom of the placode, whereas toll-2 emerges at the top.

2) Furthermore, the finding that Scr mutants eliminate tollo-/toll-8 exclusion doesn't provide much insight as the tissue is likely transformed and no longer able to form a salivary gland placode. If the placode forms, then the author should provide supporting evidence.

3) Would the authors expect that overexpression of any Toll gene would perturb the Toll signaling code/morphogenesis of the salivary gland? And if that is the case, are the phenotypes arising from ectopic expression of Tollo that surprising? The reason this is important, in my opinion, is because Tollo may simply be a gene that is associated with development of "non-salivary gland epidermal cells". Is it really surprising that as cells within the placode develop - some become salivary gland duct cells, others become secretory cells, and yet others specify epidermal cells?

4) The reviewer would caution the authors to explore the relationship between clusters identified in Fig. 2A before focusing then in Fig 3 onwards exclusively only on a subset of the clusters. In particular, the trajectory analysis in Fig 3E suggests that cells previously characterized as the "non-salivary gland epidermal cells" (Fig 2A) may correspond to a progenitor population that relates to salivary gland development as well as to that of the epidermal cells. For instance, in Fig 3E, purple color is found both to the right (closer to salivary gland cells) as well as to the left (closer to the enhancer of split-enriched cells).

5) The authors should add fkh and gfp expression to the dot plot in Fig 3B. Is fkh expressed by the majority of cells in the cluster? gfp would serve as a nice reference control.

6) The authors should show a feature UMAP plot, displaying individual cells within the UMAP associated with Tollo expression. In particular, Tollo expression in the UMAP of Fig 3A and Fig 2A should be displayed, so that a comparison to "early salivary gland" and "non-salivary gland epidermal cells" clusters can be made. Again, the reviewer's concern is that Tollo is simply a gene that is expressed in other epithelial cells that will not turn into salivary glands (e.g. Enhancer of split cluster cells?). By incorporating a discussion of the potential alternative trajectory towards whatever fate the Enhancer-of-split cluster cells form, the paper might be clearer. It would not detract from the trajectory analysis of Fig 3E, to label "early" as a progenitor population.

7) With regard to trajectory analysis - does duct form before secretory cells? If not then can the trajectory analysis support the pseudotime model? In Fig 3F, the genes bnl, sano, fkh, fog, Gmap, Fili, nur, ogre, Calr appear to support a temporal progression; however dynamic expression is only shown for Gmap in Fig 3D. In other words, the dynamic expression may pertain to only this particular subset of genes. The other marker genes (Tollo, hth, CG4528) might represent other cell types - not "earlier" salivary gland cells. If this is the case, then the exclusion of Tollo has to be rephrased as a cell fate decision. It would also help to show dynamics associated with genes besides Gmap such at bnl, sano, fog, fili etc.

8) The expression of fkh-Gal4 is shown in Fig 1A using anti-GFP staining is confusing. No expression is detected at stage 10 and yet the authors claim that cells of stage 10 (gland specification?) were identified. How can this be the case? Is the staining lacking sensitivity? And if that is the case, were cells outside the salivary gland placode expressing GFP? Which cells are these?

9) How were the scRNA-seq data processed to control for doublets?How many cells were eliminated and what steps were involved.

10) The datasets should be deposited to GEO at the time of review.

Minor points:

11) Fig 2C - It's unclear what comparisons were made to calculate the fold change. Is the orange box "early vs early" whereas the blue box is "specified duct vs early" and the green is "specified secretory vs specified duct"? In general, attention should be paid to providing more information in the figures and/or legends.

12) Fig 4C - what comparison is made here? E(spl)-enriched cluster relative to what? Early? Is this evidence for two roads that progenitor cells might take… salary gland versus E(spl)-expressing epidermal cells?

Reviewer #3: This work by May and Röper uses single-cell RNA sequencing to investigate the genetic regulation of salivary gland (SG) morphogenesis in Drosophila. The authors establish an improved protocol to isolate SG cells during a specific time window, based on comparison with epidermal cells, enriching the desired cell population for analysis. This is crucial as SG cells are a small fraction (less than 1%) of embryonic cells, and past sc-RNAseq methods struggled to identify them.

The authors describe upregulated SG genes compared to the epidermis and assess dynamic expression changes in SG clusters, revealing tightly regulated temporal patterns of gene expression. HCR in-situ hybridization confirms several identified genes. They identify genes excluded from the secretory placode during SG development, focusing on Tollo's exclusion. Continued Tollo presence leads to unregulated apical constriction and invagination defects, associated with abnormal actomyosin accumulation. The study further concludes that Tollo regulation is downstream of Scr.

Albeit mainly descriptive, this work is important, and the data analysis reveals interesting genetic data. The work gives an overview of the dynamic genetic control of SG morphogenesis and will be of significance to the fields of morphogenesis and organogenesis.

Major comments:

1 - A recent study by sc-RNAseq analysed the transcriptomes of Drosophila embryos at two different stages and succeeded in identifying SG cell types and transcripts (Peng et al., Development 2024, doi:10.1242/dev.202097). These results should be mentioned, and it would be interesting to compare them with the current data. Particularly since the work by May and Roeper seems more extensive and SG specific.

2 - Expression of CG45263 is not clear from the panels on figure 3 and figure S3, where the expression seems to be restricted to the nervous system. These panels should be redone to clearly show what the authors describe in the text.

Minor comments:

The authors should explain the rationale for using Tollo without a cytoplasmatic domain (UAS-TolloΔcyto-GFP).

Reviewer #4: PBIOLOGY-D-24-01652

In this paper, May and Röper investigated the gene expression profiles in salivary gland and ectodermal cells by performing single-cell RNA sequencing analysis. The authors purified fkh> Gal4-expressing salivary gland cells and whole ectodermal cells (including salivary gland) expressing the pan-ectodermal gene arm using FACS sorting, an approach different from the most of other scRNAseq works analyzing cells from whole embryos. Cell data from the two sources are combined and classified into multiple groups. The authors identified a group of genes with low expression in the salivary gland compared to the ectoderm. One of the genes was tollo/toll8, a leucine-rich repeat membrane protein involved in cell sorting. Forced expression of tollo in the salivary gland interfered with normal morphogenesis.

Two major points are made. One is the exclusion of a subset of genes expressed in the early stage of salivary gland development. The other is the significance of tollo exclusion in salivary gland development. This reviewer found uncertainty in both claims, which the authors must address. Given the weakness in the exclusion model and some uncertainty in the presentation of the data, this reviewer is not convinced of the magnitude of the progress made in this version of the manuscript.

Major point.

1. Handling and interpretation of single-cell RNA sequencing data.

1-1 If I understand correctly, the authors obtained scRNAseq data of purified fkh>GFP+ cells and arm-GFP cells and combined the two data sets before clustering. Then, each cluster (Fig. 3A, Fig. 4A) contains cells of fkh>GFP and arm-GFP fractions. Information on the contribution from each FACS sorted fraction would help interpret the data. This will also provide information on the purity of FACS sorting.

1-2 The main claim derived from the analysis shown in Fig. 2 is that the collected cells are classified into five clusters representing the temporal order of salivary gland development (early, specified duct, specified secretory and post specification). I repeat the question of 1-1: To what extent the two FACS sorted cell fractions contribute to each cluster? Some of the none-salivary gland cell clusters (early and specified duct) may include more cells from the arm-GFP fractions, which should be derived from the broad ectodermal territories not covering the PS2. Then how did the author justify their naming?

2. The strong point made in this manuscript is the identification of some genes excluded from the salivary gland to allow gland development. This class of gene includes grh, hth and tollo/toll8.

2-1 Regarding the "non-salivary gland epidermal cells " The authors presented tollo expression (Fig. 4E, 6AB), in which the authors described the gene as expressed at the time of salivary gland placode specification (page 11, paragraph 1). I am not convinced with this view of tollo "exclusion". The tollo gene seems simply not expressed in the placode: no solid evidence for active exclusion of tollo expression from the salivary gland placode is presented. "Exclusion" is a strong word. It should be presented with robust evidence. Although the ectopic expression of tollo interfered with the salivary gland development, such an effect has been described in other cases (ex. https://doi.org/10.1002/dvg.20245), and its implication is limited.

2-2 Down-regulation of grh and hth was clearly documented (Fig. 4E, S3A). But their functional implication is not known.

Other comment.

3. Log2 fold change values should be clearly defined. In their second reply to my inquiry, the authors presented a table listing group 1 and 2 cells for the log2 fold change value calculation. The choice of groups 1 and 2 differed from one data set to another. That description was totally absent in the original manuscript and made it impossible to evaluate the data. In the table, I believe "Figure 2C" should be "Figure 3C". Is the description for group 2 cells described as "Remaining cells in the dataset" for the description of Figure 2C (which I believe is Figure 3C) available at any place in the manuscript?

4. Page 8. 2nd paragraph. "pip (pipe), a sulfotransferase of the Golgi is key to the later secretion function of the salivary glands and a known marker of these, and its transcript colocalised already early on with fkh mRNA, confirming this cluster as 'salivary gland' (Fig. 2B, B'). "

Misleading logic. This data confirmed that one gene enriched in this cluster, pipe, is expressed in the salivary gland. If the authors intend to claim this cluster as a 'salivary gland', the expression of multiple genes should be tested before making a firm conclusion.

5. Page 8. last paragraph. "Thus, the single cell RNA-sequencing analysis of cells marked by and isolated based on GFP expression under fkhGal4 control (UAS-srcGFP fkhGal4) was able to generate a cell atlas of the salivary gland placode as well as its precursor epidermis and nearby tissues at early stages of embryogenesis that provides a rich resource of expression data for these stages. "

This reviewer is not convinced with this argument. The muscle, CNS, hemocyte clusters may be simply a contamination. Other possibility is that additional genes are activated during cell dissociation and FACS sorting. It was reported that Notch signaling is activated by dissociation-dependent ectodomain shedding of Notch receptor. The E(spl) positive clusters may be such an example.

6. Page 11, para1, "At the gland specification stage, both tollo and grh were still expressed in parasegment 2 where the salivary gland placode will form, but both were clearly excluded from the secretory part of the placode once apical constriction commenced. "

(This comment overlaps with comment 2-1). I agree with the authors' view that grh is initially expressed in the salivary gland placode and down-regulated afterward. But I disagree that tollo is expressed in the salivary gland placode (left most image). There is a signal in the placode, but the level seems to be the same range in the later stage. The tollo expression pattern shown in Fig. 5A indicates this gene is expressed in a striped pattern outside of the sg placode. Without additional expression study on the earlier stage with higher tollo expression, the statement of tollo down-regulation is not supported.

Is the low tollo expression in the placode is due to the salivary grand program? The authors provided a data to tollo expression in Scr mutant (Fig. 6D), which showed slight upregulation. Since Scr is involved in a broader program of segment identity, implication of the result should be carefully evaluated. I feel this level of tollo expression change is not sufficiently strong to justify the "exclusion" model. The authors also showed that tollo expression does not change (upregulated) in fkh mutants (Sup Fig. 4C), arguing against the exclusion model.

7. Staging description should be consistent. The text sometimes followed the staging of Campos-Ortega and Hartenstein, in other place "start of gland specification", "apical constriction", etc are used.

8. Typo. Page 7 line 17. 17 genes we highly upregulated -> 17 genes were highly upregulated

9. Fig. 2B-G. Please specify the stage of embryos.

---

## [Decision Letter · Decision Letter 2]

11 Mar 2025

Dear Dr Röper,

Thank you for your patience while we considered your revised manuscript "Single cell transcriptomics of the Drosophila embryonic salivary gland reveals not only induction but also exclusion of expression as key morphogenetic control steps" for consideration as a Research Article at PLOS Biology. Your revised study has now been evaluated by the PLOS Biology editors, the Academic Editor and the original reviewers.

The reviews are appended below. As you will see, the reviewers are largely satisfied by the revision but reviewers 2 and 4 flag a couple of last minor issues that we think should be addressed in another revision. We anticipate the next revision should not take you very long. We will then assess your revised manuscript and your response to the reviewers' comments with our Academic Editor aiming to avoid further rounds of peer-review, although we might need to consult with the reviewers, depending on the nature of the revisions.

**IMPORTANT: As you address the last reviewer comments, please also address the following editorial requests:

1 - TITLE: We would like to propose a small tweak to the title. If you agree, we suggest that you change it to:

Single-cell analysis of the early Drosophila salivary gland reveals that morphogenesis is regulated by both the induction and exclusion of gene expression programs.

2 - FINANCIAL DISCLOSURES: Please take a moment to update your financial disclosures statement in our online system. A complete financial disclosures statement will include the following details:

Initials of the authors who received each award

Grant numbers awarded to each author

The full name of each funder

URL of each funder website

A statement indicating whether the sponsors or funders played any role in the study design, data collection and analysis, decision to publish, or preparation of the manuscript

3 - DATA: thank you for depositing your RNA-seq data on GEO. I see that this is currently private - which is OK for now, but please note that this will need to be made public before publication.

4 - DATA: For figures containing flow cytometry data data, we ask that you provide FCS files and a picture showing the successive plots and gates that were applied to the FCS files to generate the figure. As these files are usually quite big, you can deposit them in the Flow Repository (http://flowrepository.org/) and please make sure they are publicly available.

I have heard that Flow Repository has been giving some AUs issues lately - and if you find that the case, you can use a different repository.

5 - DATA: In addition to providing the raw RNA-seq and FCS files, detailed above, to be fully compliant with PLOS' Data Policy we also require that all other data be made available without restriction: http://journals.plos.org/plosbiology/s/data-availability. For more information, please also see this editorial: http://dx.doi.org/10.1371/journal.pbio.1001797

a. Supplementary files (e.g., excel). Please ensure that all data files are uploaded as 'Supporting Information' and are invariably referred to (in the manuscript, figure legends, and the Description field when uploading your files) using the following format verbatim: S1 Data, S2 Data, etc. Multiple panels of a single or even several figures can be included as multiple sheets in one excel file that is saved using exactly the following convention: S1_Data.xlsx (using an underscore).

b. Deposition in a publicly available repository. Please also provide the accession code or a reviewer link so that we may view your data before publication.

>>Regardless of the method selected, please ensure that you provide the individual numerical values that underlie the summary data displayed in the following figure panels as they are essential for readers to assess your analysis and to reproduce it:

Fig 5D-E

>>Please also ensure that figure legends in your manuscript include information on where the underlying data can be found, and ensure your supplemental data file/s has a legend.

>>Please ensure that your Data Statement in the submission system accurately describes where your data can be found.

6 - CODE: Thank yo for providing the scripts used here on Github. Per journal policy, for sharing custom code during the course of this investigation, we cannot accept sole deposition of code in GitHub, as this could be changed after publication. However, you can archive this version of your publicly available GitHub code to Zenodo. Once you do this, it will generate a DOI number, which you will need to provide in the Data Accessibility Statement (you are welcome to also provide the GitHub access information). See the process for doing this here: https://docs.github.com/en/repositories/archiving-a-github-repository/referencing-and-citing-content

**IMPORTANT - SUBMITTING YOUR REVISION**

*Resubmission Checklist*

*Published Peer Review*

Sincerely,

Luke

Lucas Smith, Ph.D.

Senior Editor

PLOS Biology

lsmith@plos.org

REVIEWS:

Reviewer #1: May and Röper have addressed the various issues raised by this reviewer. Notably, they have made an effort to compare several similar studies and provide a more insightful analysis of Tollo receptors. These revisions have improved the manuscript's clarity. While the results regarding Tollo receptors remain inconclusive, they are potentially interesting. The manuscript, however, remains primarily descriptive to my opinion.

Reviewer #2: The authors have addressed my concerns, and this version is more clear and easier to follow.

The following two points are suggested edits:

Rebuttal statement: "Furthermore, with regards to potential inter-regulation of Toll expression, we have now analysed toll-

2/18w expression and also toll-6 expression when Toll-8/Tollo is overexpressed and find that there is

no change to either of the other two Tolls at the transcriptional level (see Suppl. Fig. S6)."

>I assume you are referring to Suppl. Fig. S7 instead.

Rebuttal statement: "As already explained above, our data (∆cyto expression as well as retained CrebA labelling)

demonstrate that Tollo/Toll8 is highly unlikely functioning as a cell fate switch."

>My question is not whether Tollo is driving the cell fate switch but rather that it is not surprising to detect the downregulation of certain genes as cells differentiate.

What I find helpful is the Tollo expression shown in Figure S5A, as it confirms that Tollo is expressed in a subset of cells within the cluster related to the salivary gland.

I would also encourage you to show Tollo expression in the subset of cells analyzed in Suppl. Fig. S3.

Reviewer #3: The authors have improved the manuscript and figures and have answered all my concerns.

Reviewer #4: Main Comment

The revised manuscript of May and Roper addressed most of the questions I raised for the previous version. Quantification of toll8 mRNA revealed clear down-regulation in the developing salivary gland placode. Additional analysis of the upstream transcriptional program under the control of Scr and hairy provided further insight into toll8's negative regulation.

Minor Comment

page 14 line 10 from bottom. : Should (Suppl. Fig. 4C) be Suppl. Fig. 7C?

---

## [Editor Report · Decision Letter 3]

25 Mar 2025

Dear Dr Röper,

Thank you for the submission of your revised Research Article "Single-cell analysis of the early Drosophila salivary gland reveals that morphogenetic control involves both the induction and exclusion of gene expression programs" for publication in PLOS Biology and thank you for addressing the last reviewer and editorial requests in this revision. On behalf of my colleagues and the Academic Editor, Nicolas Tapon, I am pleased to say that we can in principle accept your manuscript for publication, provided you address any remaining formatting and reporting issues. These will be detailed in an email you should receive within 2-3 business days from our colleagues in the journal operations team; no action is required from you until then. Please note that we will not be able to formally accept your manuscript and schedule it for publication until you have completed any requested changes.

PRESS

We frequently collaborate with press offices. If your institution or institutions have a press office, please notify them about your upcoming paper at this point, to enable them to help maximize its impact. If the press office is planning to promote your findings, we would be grateful if they could coordinate with biologypress@plos.org. If you have previously opted in to the early version process, we ask that you notify us immediately of any press plans so that we may opt out on your behalf.

Sincerely, 

Lucas Smith, Ph.D.

Senior Editor

PLOS Biology

lsmith@plos.org